# Micro-environmental sensing by bone marrow stroma identifies IL-6 and TGFβ1 as regulators of hematopoietic ageing

Simona Valletta[1], Alexander Thomas[1], Yiran Meng[1], Xiying Ren[1], Roy Drissen [1], Hilal Sengül[1], Cristina Di Genua[1] & Claus Nerlov [1]✉

Hematopoietic ageing involves declining erythropoiesis and lymphopoiesis, leading to frequent anaemia and decreased adaptive immunity. How intrinsic changes to the hematopoietic stem cells (HSCs), an altered microenvironment and systemic factors contribute to this process is not fully understood. Here we use bone marrow stromal cells as sensors of age-associated changes to the bone marrow microenvironment, and observe up-regulation of IL-6 and TGFβ signalling-induced gene expression in aged bone marrow stroma. Inhibition of TGFβ signalling leads to reversal of age-associated HSC platelet lineage bias, increased generation of lymphoid progenitors and rebalanced HSC lineage output in transplantation assays. In contrast, decreased erythropoiesis is not an intrinsic property of aged HSCs, but associated with decreased levels and functionality of erythroid progenitor populations, defects ameliorated by TGFβ-receptor and IL-6 inhibition, respectively. These results show that both HSC-intrinsic and -extrinsic mechanisms are involved in age-associated hematopoietic decline, and identify therapeutic targets that promote their reversal.

[1] MRC Molecular Hematology Unit, MRC Weatherall Institute of Molecular Medicine, John Radcliffe Hospital, University of Oxford, Oxford OX3 9DS, UK.
✉email: claus.nerlov@imm.ox.ac.uk

During ageing erythropoiesis and lymphopoiesis both decrease. The former leads to a significantly increased frequency of anaemia in the elderly population, which reaches 25–30% at 85 years of age[1]. Anaemia is clinically defined as low blood haemoglobin (Hb) (≤12 g/dl in women, ≤13 g/dl for men), and is associated with several key age-related morbidities, including cognitive[2], musculoskeletal[3] and cardiovascular decline, as well as higher overall mortality[4]. Decreased lymphopoiesis impairs de novo generation of naive B-cells and T-cells[5]. This leads to a declining antibody- and T-cell receptor repertoire and impaired immune response to key pathogens such as respiratory syncytial virus and influenza virus and their vaccines[6]. Understanding the underlying causes of the age-associated decrease in erythropoiesis and lymphopoiesis, and the development of pharmacological strategies to counteract them, is therefore an important step towards improving the health of the ageing population.

Hematopoietic stem cells (HSCs) maintain the hematopoietic system throughout the lifespan of the vertebrate organism. However, the HSC compartment, as well as the properties of individual HSCs, undergoes significant changes with age. Repopulating HSCs expand[7,8], while their individual ability to generate differentiated progeny declines[9,10]. In addition, HSCs are heterogeneous in their lineage output, as demonstrated by single HSC transplantation[11–14], and the proportions of fate-restricted and lineage-biased HSCs change with age: HSCs with platelet and myeloid lineage bias become more prevalent, while the frequency of HSCs that produce lymphoid cell types decreases[15–17]. The latter observation provides a cellular basis for the age-dependent decline in de novo production of naive B- and T-lymphocytes[18].

While age-dependent changes to HSC function and lineage output are well established, the underlying causes are less well understood. The functional decline of HSCs has been linked to their proliferative history, as highly quiescent HSCs retain their repopulating capacity over time[19,20]. Increased molecular and functional platelet bias of aged HSCs has been observed in both mice and humans[17,21], and HSC platelet programming may directly inhibit lymphocyte production[17]. In contrast, we do not know how ageing affects the intrinsic ability of HSCs to produce erythroid cells. This is principally due to technical limitations: identification of the cellular output from HSCs has traditionally used CD45 allotypic markers, and erythrocytes do not express CD45. Therefore, quantitative studies measuring functional differences between young and aged HSCs have thus far not addressed if their intrinsic capacity for erythropoiesis is altered. Age-dependent intrinsic changes to HSC lineage output may be influenced by extrinsic factors: Ccl5[22] and Wnt5a[23] have both been observed to promote age-dependent myeloid bias, as has the age-dependent loss of β-adrenergic innervations in the bone marrow[24], with the caveat that these studies did not measure platelet contributions from HSCs. Finally, the circulating levels of inflammatory cytokines, such as IL-6 and TNFα, increase during normal ageing in both mice[25] and humans[26], with the levels both cytokines similarly increased in bone marrow[27,28], and have been shown to inhibit erythropoiesis both in vitro[28] and in vivo[29]. However, their impact on HSC and downstream progenitor function during physiological ageing remains to be experimentally determined.

We here use comprehensive readout of hematopoietic lineage in competitive repopulation assays to show that reduced lymphoid, but not erythroid, lineage output is an intrinsic property of aged HSCs. By gene expression profiling of multiple stromal populations from young and aged bone marrow and pathway analysis we identify TGFβ and IL-6 signalling as increased in the aged micro-environment. Inhibition of IL-6 signalling in aged mice using a neutralising anti-IL-6 antibody reverted the age-associated decrease in erythroid progenitor activity. Inhibition of TGFβ signalling rebalanced the lineage output from aged repopulating HSCs and increased their lymphoid progenitor production, and in addition ameliorated the decrease in erythroid progenitor numbers seen in aged mice. Therefore, inhibition of IL-6 and TGFβ signalling in aged mice co-operatively counteract the age-associated decrease in erythroid and lymphoid cell production.

## Results

**Decreased erythroid output is not intrinsic to aged HSCs.** As observed during human ageing, aged mice have lower haematocrit (HCT) and Hb levels compared to young mice, whereas their platelet count is increased (Fig. 1a). To determine whether decreased erythropoiesis is due to an intrinsic defect of aged HSCs we transplanted phenotypic long-term (LT-)HSCs (Lin⁻c-Kit⁺Sca-1⁺CD150⁺CD48⁻CD34⁻)[30,31] (Fig. 1b) from young (2–3-month-old) and aged mice (24-month-old) carrying the Vwf-tdTomato and Gata1-EGFP transgenes (VT/GE mice), which allow for accurate readout of erythrocytes, which are labelled by EGFP⁺, as well as platelets, which are both EGFP⁺ and tdTomato⁺[14]. Fifty VT/GE long-term HSCs (CD45.2 allotype) were transplanted into young CD45.1 recipients along with 200,000 young wild-type CD45.1 competitor cells (Fig. 1c) and long-term reconstitution of all hematopoietic lineages readout using CD45 allotypes (Fig. 1d) and the Vwf-tdTomato and Gata1-EGFP transgenes (Fig. 1e, f). As previously reported, aged HSCs show an overall lower level of lineage reconstitution, with a particularly prominent decrease in lymphocyte output:[8,9,16,17] while for platelets, erythrocytes and myeloid cells a 1.5–2-fold decrease in reconstitution was seen, this difference was fourfold to fivefold for B-cells and T-cells (Fig. 1g). Since the overall number of repopulating HSCs increases with age, compensating for the decrease in lineage output on a per cell basis, the critical issue is their relative contribution to hematopoietic lineages. Normalisation to platelet output, which is the lineage with the highest output from both young and aged HSCs, as well as single fate-restricted HSCs[14], showed that lymphoid output from aged HSCs was decreased, whereas myeloid and erythroid-lineage output did not decline relative to platelet output during ageing (Fig. 1h). Decreased erythroid-lineage output is therefore not an intrinsic property of aged HSCs, whereas decreased lymphoid cell output is.

**The function of erythroid progenitors decrease with age.** To address whether specification or differentiation of erythroid-lineage progenitors was impaired in an aged bone marrow environment, we compared the number and function of hematopoietic stem and progenitor cells between young and aged mice. As previously reported, during ageing the frequency of LT-HSCs and multi-potent (LSK) progenitor cells increases (Fig. 2a, b). We also observed an increase in total Lin⁻c-Kit⁺ (LK) myelo-erythroid progenitors (Fig. 2c). Analysis of the LK compartment[32] (Fig. 2d) showed an increase in megakaryocyte (MkP) and myeloid (GMP) progenitors, whereas the frequency of erythroid-lineage progenitors (preMegE, preCFU-E, CFU-E) (Fig. 2e) was significantly decreased. To measure the functional state of erythroid progenitors we purified MPP2s, preCFU-Es and CD71⁺ CFU-Es from young and aged mice and measured their colony-forming capacity, observing a decrease in clonogenic erythroid progenitors for all three populations during ageing (Fig. 2f). Gene expression profiling and gene set enrichment analysis (GSEA) of preCFU-Es showed that expression of genes normally associated with erythroid-lineage specification was significantly decreased in old compared to young mice (Fig. 2g), with a

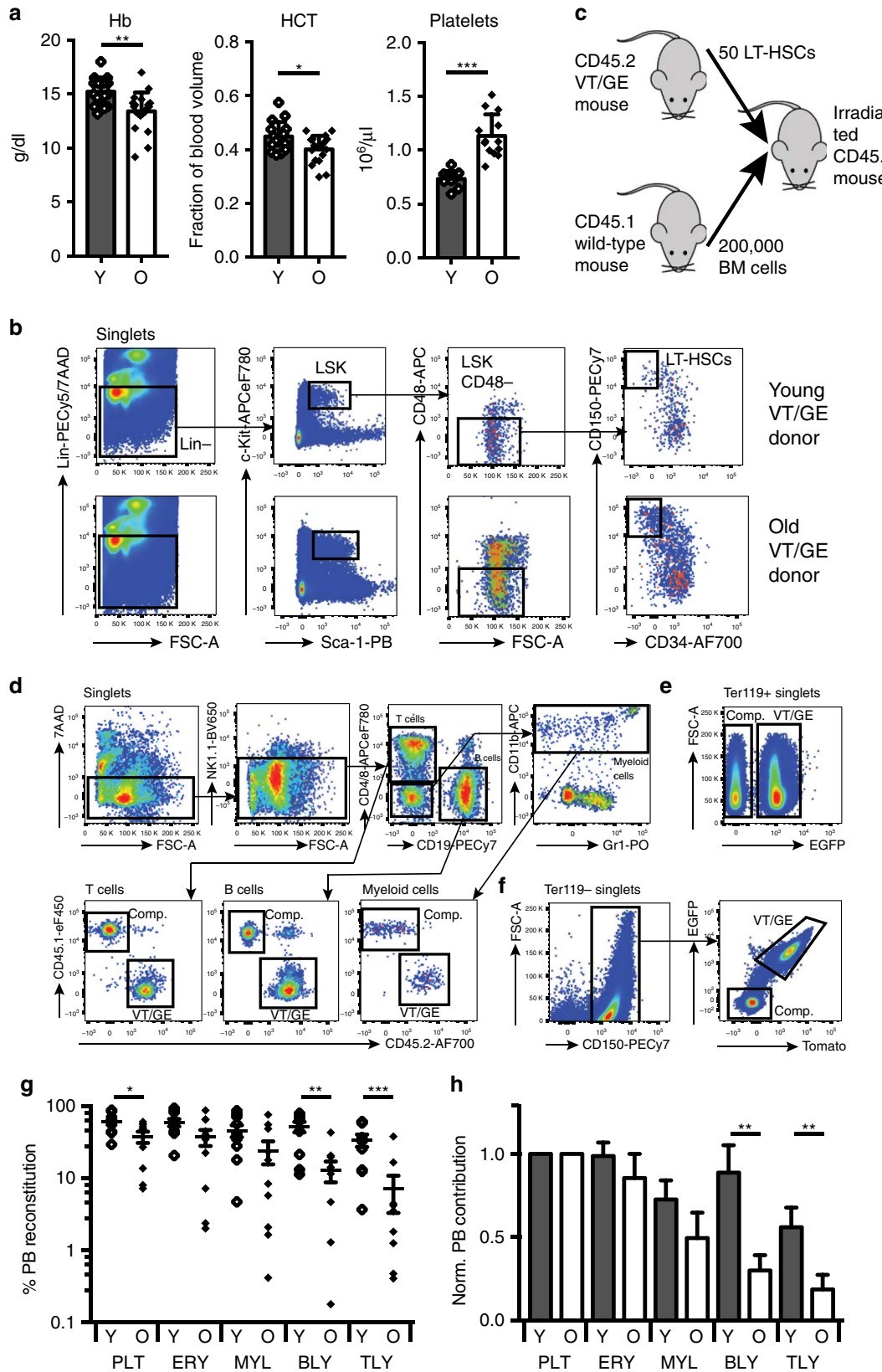

corresponding increase in preGM-associated gene expression (Fig. 2h). Accordingly, we observed that the aged preCFU-E and MPP2 populations, in addition to their lower erythroid output, also displayed a significant increase in myeloid colony formation (Fig. 2i).

**Identification of deregulated signalling in aged bone marrow.** The bone marrow micro-environment is under the combined influence of locally and systemically produced factors. To assess the overall effect of ageing on the bone marrow micro-environment at the molecular level, we performed RNA sequencing of sorted bone

**Fig. 1 Effect of ageing on erythropoiesis. a** PB haemoglobin (Hb), haematocrit (HCT) and platelets number measured in young (red cell parameters: $N = 16$; platelets: $N = 11$) and old mice (red cell parameters $N = 20$; platelets: $N = 13$). Data are from seven independent experiments. Values show mean ± s.e.m. *$P < 0.05$; **$P < 0.01$; ***$P < 0.001$ (two-tailed, unpaired Student's $t$ test). Exact $P$ values: Hb: 0.0014; HCT: 0.026; Platelets: 0.000003. **b** Gating strategy for sorting of LT-HSCs from BM of young and old VT/GE CD45.2 transgenic mice used as donors for competitive transplantation. **c** Experimental design to compare in vivo lineage output of transplanted young and old LT-HSCs. Strategy for measuring donor PB leucocyte (**d**), erythrocyte (**e**) and platelet reconstitution (**f**) in transplanted recipient mice. **g** PB lineage contribution of young and old HSCs in transplanted recipients ($N = 10$/condition). The mean values ± s.e.m are indicated, as are $P$ values (two-sided Mann–Whitney $U$-test) comparing young and old contributions for each lineage. *$P < 0.05$; **$P < 0.01$; ***$P < 0.001$. Exact $P$ values: PLT: 0.03; ERY: 0.12; MYL: 0.06; BLY: 0.003; TLY: 0.003. **h** Donor HSC PB contribution from **g** normalised to platelet output. The mean values ± s.e.m are indicated, as are $P$ values (two-sided Mann–Whitney $U$-test) comparing young and old contributions for each lineage. **$P <$ 0.01. Exact $P$ values: PLT: not applicable; ERY: 0.35; MYL: 0.17; BLY: 0.007; TLY: 0.009). Source data are provided as a source data file.

lining (BL) osteoblasts (OBs: CD45⁻Ter119⁻ALCAM⁺), mesench- ymal stromal cells (MSCs: CD45⁻Ter119⁻CD31⁻ALCAM⁻Sca-1⁺ PDGFRα⁺) and vascular endothelial cells (VECs: CD45⁻Ter119⁻ CD31⁺), as well as central bone marrow (CBM) VECs, LepR⁺ PDGFRα⁺ and LepR⁺PDGFRα⁻ perivascular cells (PV⁺; CD45⁻ Ter119⁻CD31⁻LepR⁺PDGFRα⁺, PV⁻; CD45⁻Ter119⁻CD31⁻ LepR⁺PDGFRα⁻) (Supplementary Fig. 1a, b). The identity of the sorted cell populations was validated by analysis of the genes encoding their characteristic markers (Supplementary Fig. 1c). We observed that a large number of genes were regulated by age in all cell types (Fig. 3a and Supplementary Data 1). To identify signalling pathways underlying these gene expression changes we performed Metacore process network analysis, focusing on process networks that were enriched during ageing across multiple aged stromal cell types, as these were likely to reflect general changes to the micro- environment, rather than effects of ageing specific to individual cell types. This analysis revealed that several inflammatory process networks were enriched in the majority of aged BM stromal cell types, most prominently those associated with IL-2, IL-6 and IL-10 signalling (Fig. 3b and Supplementary Data 2). *Il2ra* was not expressed on stromal cells, and *Il10ra* at low levels and only in aged OBs and CBM-VECs (Supplementary Fig. 2a, b). In contrast *Il6st* was expressed at high levels in aged stromal cell types where the IL-6 network was highly enriched, and at low levels in those showing weaker enrichment (Supplementary Fig. 2c). Finally, GSEA showed that IL-6-induced gene expression was significantly enriched in aged mice in the same stromal cell types that showed a high IL-6 process enrichment and high *Il6st* expression (Supplementary Fig. 2d).

**IL-6 inhibition improved aged erythroid progenitor function.** These observations were consistent with elevated IL-6 signalling in the aged bone marrow environment contributing to the age- dependent decline in erythropoiesis. We therefore treated aged mice with a neutralising anti-IL-6 antibody or control IgG for 3 weeks and characterised their erythroid progenitor compart- ment both phenotypically and functionally. Treatment with IL-6 antibody did not affect the number of phenotypic HSCs, LSK or LK cells (Supplementary Fig. 3a–c). Analysis of the myelo- erythroid progenitor compartment showed a small, but sig- nificant, increase in phenotypic preMegEs and a similar decrease in GMPs, consistent with IL-6 contributing to myeloid skewing of progenitor differentiation. However, we did not observe any change in the proportion of phenotypic committed erythroid progenitor cells upon IL-6 inhibition (Fig. 3c). To address whe- ther inhibition of IL-6 affected erythroid or myeloid progenitor function we cultured LK cells isolated from anti-IL-6- and control IgG-treated mice under erythroid and myeloid conditions. We observed a highly significant increase in erythroid colony-forming cells (Fig. 3d), and a corresponding decrease in myeloid colony- forming cells (Fig. 3e). Also, the erythroid colony-forming activity of purified preCFU-Es was increased (Fig. 3f). GSEA analysis of anti-IL-6- and IgG-treated preCFU-Es showed that age-associated

depletion of preCFU-E-specific gene expression was observed in preCFU-Es from IgG-treated, but not anti-IL-6-treated, mice (Fig. 3g, h), indicating that IL-6 inhibition improves erythroid progenitor function by restoring erythroid-specific gene expres- sion in progenitors. This analysis also showed a lesser degree of upregulation of the preGM signature in preCFU-Es from anti-IL- 6-treated aged mice compared to control IgG-treated aged mice; however, this difference was not significant (Supplementary Fig. 3d, e). We did not observe increased expression of *Il6* during ageing in any of the stromal cell populations examined (Supple- mentary Fig. 3f), or change to their expression of key erythroid cytokines (*Kitl*, *Epo*; Supplementary Fig. 3g, h). This would be consistent with increased levels of circulating IL-6, rather than altered local production of supporting cytokines in the BM, being responsible for the age-dependent activation of inflammatory signalling pathways in bone marrow stromal cells, and the asso- ciated decline in erythropoiesis. To determine the effect of ele- vated systemic IL-6 levels we overexpressed IL-6 in young mice using hydrodynamic injection, as previously described[33]. We observed that increased circulating IL-6 (Supplementary Fig. 3i) led to an increase in GMPs and a decrease in preMegEs (Sup- plementary Fig. 3j), the converse effect of IL-6 inhibition in aged mice, substantiating a direct role of elevated circulating IL-6 levels in the aged progenitor phenotype.

**TGFβR1 inhibition reduces HSC number in aged mice.** While IL-6 inhibition improved erythroid progenitor function, it did not restore the number of phenotypic erythroid progenitors in aged mice, nor did it counteract the increase in MkPs that is associated with age-dependent HSC platelet bias[17]. In the Metacore analysis the TGFβ/GDF/activin process network was enriched in 3/6 aged stromal cell populations. Inhibition of TGFβ and activin signal- ling has been shown to ameliorate ineffective erythropoiesis in myelodysplastic syndrome[34] and β-thalassaemias[35], respectively. In addition, loss of HSC quiescence was observed after mega- karyocyte (MK)-specific ablation of *Tgfb1*[36], and MKs are enri- ched in the niches of Vwf⁺ platelet-biased HSCs[37]. Since platelet- biased HSCs are preferentially expanded during ageing[17] increased TGFβ signalling therefore had the potential to both impair erythropoiesis and promote HSC platelet bias. We examined the expression of TGFβ family genes encoding between young and aged stromal cell types (Supplementary Fig. 4a). This showed that *Tgfb1* expression was strongly and consistently upregulated in aged stromal cells (Supplementary Fig. 4b), while no such upregulation was observed for other known extrinsic regulators of HSC function (*Kitl*, *Cxcl12*, *Angpt1*) (Supplementary Figs. 3g and 4c, d). Consistent with these observations, the level of TGFβ1 protein was elevated in the bone marrow plasma of old compared to young mice (Supplementary Fig. 4e). We therefore treated aged mice with SB-525334 (SB), a selective inhibitor of TGFβR1[38], the catalytic subunit of the TGFβ receptor[39]. Exam- ination of HSCs and progenitors after 2 weeks of treatment

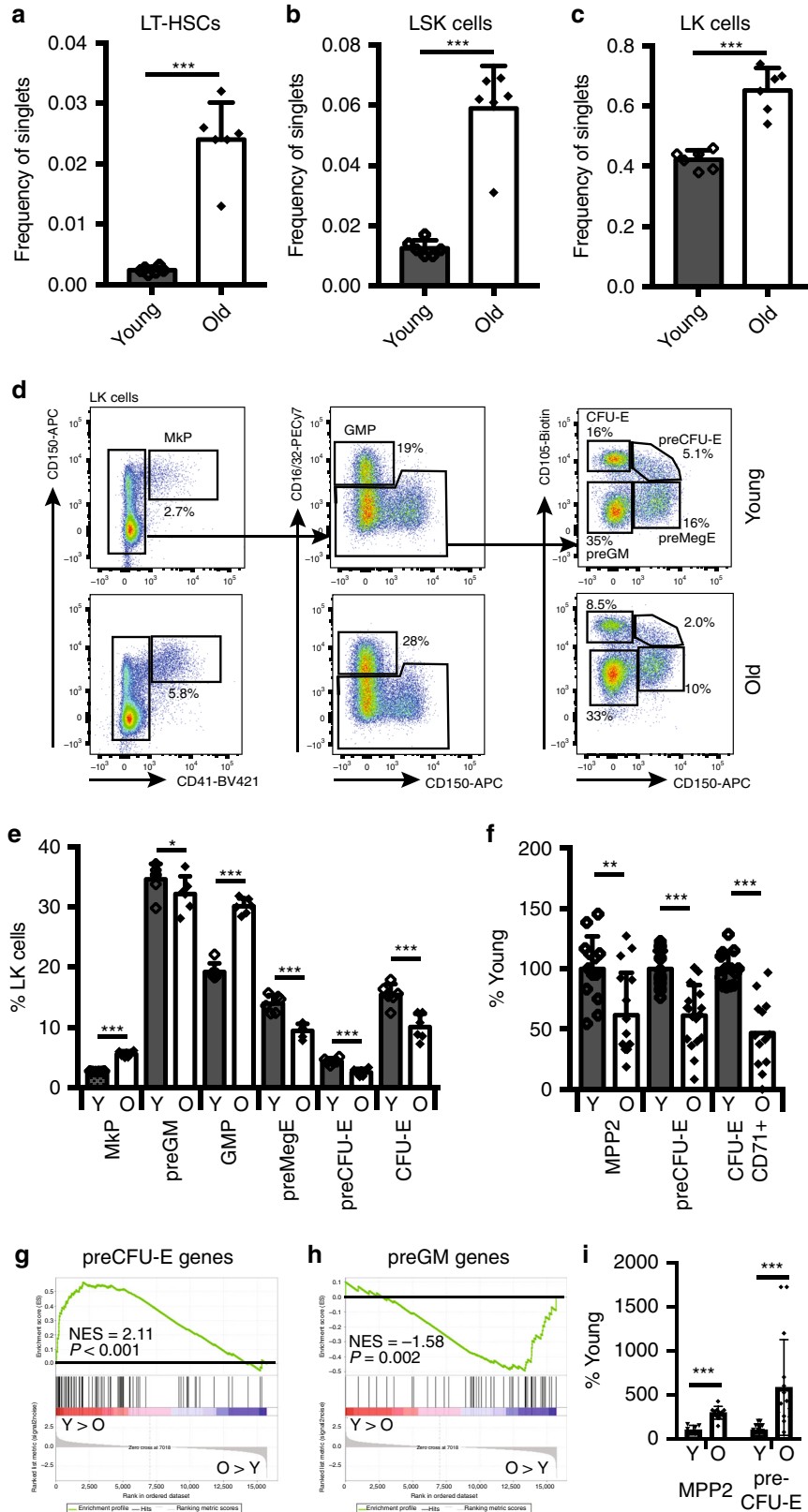

showed a highly significant decrease in phenotypic LT-HSCs (Fig. 4a), as well as a lower frequency of LSK and LK cells (Fig. 4b, c). The decrease in phenotypic LT-HSCs was accompanied by a loss of LT-HSC quiescence (Fig. 4d, e). In addition, SB treatment reduced the proportion of MkPs and GMPs, whereas preMegEs, preCFU-Es and CFU-Es were all increased (Fig. 4f, g). Finally,

the level of circulating platelets was decreased in aged mice upon SB treatment (Fig. 4h). Therefore, inhibition of TGFβR1 signalling counteracted the observed effects of ageing on the HSC, LSK and myelo-erythroid progenitor compartments, and in particular decreased MK/platelet and increased erythroid progenitor output.

**Fig. 2 Effect of ageing on HSCs and myeloid progenitors.** Bar graph showing the number of LT-HSCs (**a**), LSK (**b**) and LK (**c**) cells in bone marrow of young and old mice as frequency of live single cells ($N = 6$/condition, six independent experiments). Values are mean ± s.d. ***$P < 0.001$ (two-tailed unpaired Student's $t$ test). Exact $P$ values: LT-HSCs: 0.000007; LSK: 0.000013; LK: 0.000039. **d** Representative flow cytometry plots of myelo-erythroid progenitors from young (top panels) and old mice (bottom panels). **e** Quantification of myelo-erythroid progenitors from young and old mice ($N = 6$/condition, six independent experiments). Values are mean ± s.d. *$P < 0.05$; **$P < 0.01$; ***$P < 0.001$ (two-tailed unpaired Student's $t$ test). Exact $P$ values: MkP: 0.00000006; preGM: 0.0407; GMP: 0.00000008; preMegE: 0.0001; preCFU-E: 0.0002; CFU-E: 0.0010. **f** In vitro erythroid colony-forming potential of MPP2 (young $N = 13$, old $N = 16$), preCFU-E (young $N = 17$, old $N = 18$) and CD71+ CFU-E (young $N = 13$, old $N = 13$) cells from BM of young or old mice (six independent experiments) normalised to colony output from young mice (=100%). Values are mean ± s.e.m. **$P < 0.01$; ***$P < 0.001$ (two-tailed unpaired Student's $t$ test). Exact $P$ values: MPP2: 0.0032; preCFU-E: 0.000005; CFU-E CD71+: 0.000003. **g** GSEA analysis comparing expression of preCFU-E-specific genes in young and old preCFU-Es. The normalised enrichment score (NES) and $P$ value are shown. **h** GSEA analysis comparing expression of preGM-specific genes in young and old preCFU-Es. The normalised enrichment score (NES) and $P$ value are shown. **i** In vitro myeloid colony-forming potential of MPP2 cells (young: $N = 7$, old: $N = 10$; two independent experiments) and preCFU-E cells ($N = 6$/condition, six independent experiments) from BM of young and old mice normalised to colony output from young mice (=100%). Values are mean ± s.e.m. ***$P < 0.001$ (two-tailed unpaired Student's $t$ test). Exact $P$ values: MPP2: 0.000022; preCFU-E: 0.0009. Source data are provided as a source data file.

**TGFβR1 inhibition rebalances aged HSC lineage output.** Given the link between increased HSC platelet bias and decreased lymphopoiesis during ageing[17] we investigated whether TGFβ signalling played a role in the age-dependent decrease in lymphoid output. First, we vehicle or SB-treated aged VT/GE mice and performed single cell RNA sequencing of the LSK compartment. Cells were tSNE clustered and their expression of signatures associated with platelet-erythroid (MPP2) and lymphoid-primed MPPs (MPP4)[40] analysed (Fig. 5a, b). We observed a significant decrease in MPP2 priming, and a significant increase in MPP4 priming of LSK cells. The proportion of computationally defined MPP4 cells increased by 21% (from 26% to 32%; $P = 0.0003$; $\chi^2$-test) (Fig. 5c), and showed higher levels of MPP4-specific gene expression (Fig. 5d) after SB treatment. Consistent with this observation, culture of LSK cells from SB-treated mice showed a significant increase in B-cell output when cultured on OP9 stroma (Fig. 5e). This correlated with an increase in the number of clonogenic B-cell progenitors (Fig. 5f), whereas the response of B-cell progenitors to IL-7 was not altered after SB treatment (Supplementary Fig. 4f). To test whether increased lymphopoiesis was due to intrinsic reprogramming of repopulating HSCs we measured the lineage output of transplanted LT-HSCs from vehicle- and SB-treated VT/GE mice. We observed a significant decrease in platelet, erythroid and myeloid output from aged control HSCs, whereas lymphoid output was not affected (Fig. 5g). This led to a general rebalancing of HSC lineage output after SB treatment with highly significant increases in relative B- and T-cell output (Fig. 5h). Overall, whereas aged control HSCs showed clear platelet-lineage bias, output from SB-treated aged HSCs was similar across platelet, erythroid, myeloid and lymphoid lineages, the only significant difference being lower T-cell output (Fig. 5h).

## Discussion

We have here used bone marrow stromal cells as unbiased sensors to detect changes to the bone marrow micro-environment during ageing. This approach allowed us to identify molecular signatures associated with IL-6 and TGFβ signalling as upregulated across multiple aged stromal cell types. These results are consistent with previous observations from global molecular profiling of HSCs, which has shown upregulation of inflammation-associated genes in aged HSCs[10], as well as de-regulation of the TGFβ signalling network[41], and with the finding that levels of IL-6[27] and TGFβ (this study) are elevated in the bone marrow of aged mice. Therefore, while it is important to note that intrinsic changes to the cell types investigated, including increased expression of *Tgfb1* in aged stromal cells and age-dependent changes to the composition of the HSC compartment, have the potential to influence population-based gene profiling,

these observations clearly indicate that signalling through the IL-6 and TGFβ pathways is increased in both hematopoietic and stromal cells in the aged murine bone marrow as a result of increased ligand availability, and identify profiling of stromal cell types as a generally applicable methodology for the identification of micro-environmental changes associated with developmental or pathological processes.

In accordance with previous studies we find that ageing is accompanied by increased levels of MkP and GMP, as well as decreased levels of erythroid and lymphoid progenitor cells[8,17,21]. In addition, we show that aged erythroid progenitors are functionally impaired, correlating with a decrease in their expression of erythroid-specific genes. In humans decreased numbers of erythroid colony-forming cells have been observed in anaemic individuals[42], similarly to what is observed here in aged mice. In a separate study, serum levels of IL-6 were inversely correlated to blood Hb in frail, anaemic individuals[43]. We here observe that antibody-mediated in vivo inhibition of IL-6 leads to recovery of erythroid progenitor function, measured as colony-forming ability of the MMP2, preCFU-E and CD71+ erythroblast populations. In conjunction with our finding that aged HSCs are not impaired in their ability to generate erythrocytes when transplanted into a young BM micro-environment this supports direct inhibition of erythropoiesis by the aged micro-environment as the key component of the age-dependent decline of erythropoiesis, with IL-6 as a putative target for therapeutic intervention. However, while IL-6 inhibition improved the function of aged erythroid progenitors it did not increase their number, indicating that additional micro-environmental factors also play a role. TGFβ1 is likely to contribute, as TGFβR1 inhibition increased the number of preMegE, preCFU-E and CFU-E progenitors in aged mice. Inhibition of the TGFβ and Activin signalling improves inefficient erythropoiesis in pathological settings[34,35], and our results suggest that this concept may be extended to physiological ageing.

During ageing the hematopoietic system acquires platelet bias, associated with increased MkP and platelet levels. A key driver of this process is a >50-fold expansion of the number of platelet-primed, *Vwf*+ LT-HSCs, combined with their increased expression of platelet-lineage genes and functional platelet-restriction, as measured by single-cell transplantation[17]. The niches that sustain *Vwf*+ HSCs are enriched in MKs[37], and MK-derived TGFβ1 is critical to maintenance of HSC quiescence[36]. We here observe that bone marrow TGFβ1 levels increase with age, correlating with increased HSC platelet bias, and that blocking TGFβR1 signalling in aged mice leads to the concomitant loss of HSC platelet bias, decreased HSC quiescence, and declining number and function of HSCs. These observations are consistent with increased local TGFβ1 production by aged non-MK stromal

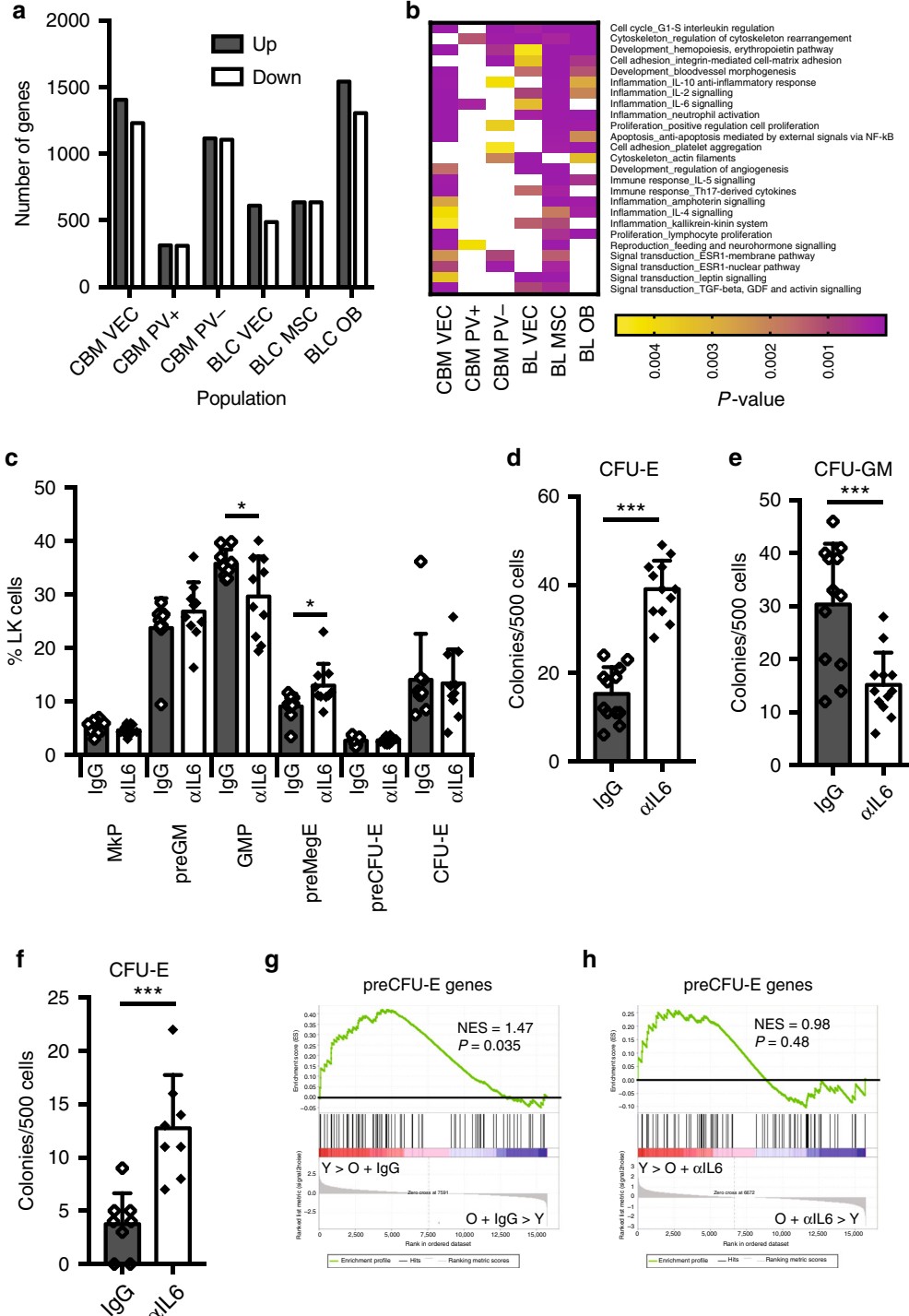

cells generating additional niches that maintain quiescent *Vwf*+ platelet-biased HSCs, and that upon TGFβR1 inhibition these niches are compromised, leading to proliferation and functional depletion of the platelet-biased HSC population. Some cells may retain their HSC immuno-phenotype after their function has been compromised, as previously observed[44–46], providing an explanation for the lower overall reconstitution of platelets, erythrocytes and myeloid cells (but not lymphocytes) by TGFβR1 inhibitor-treated HSCs.

In addition to loss of functional HSC platelet bias, TGFβR1 inhibition also improved output of lymphoid progenitors from aged HSCs. In particular, the number of clonogenic B-cell progenitors was increased threefold to 75% of the level in young mice in TGFβR1 inhibitor-treated aged mice, with a similar increase in B- and T-lymphocyte contribution to peripheral blood reconstitution after transplantation of inhibitor-treated HSCs. The functional recovery of lymphopoiesis is accompanied by downregulation of platelet-erythroid (MPP2) and upregulation of lymphoid (MPP4) gene expression in LSK cells. Overall, these results are consistent with gain of lymphoid potential being linked to loss of HSC platelet priming, as previously observed after deletion of FOG-1 from HSCs[17].

Our results show that inhibition of IL-6 and TGFβ signalling has the potential to counteract key aspects of hematopoietic

**Fig. 3 IL-6 neutralisation improves aged hematopoiesis. a** Bar graph showing the number of genes differentially expressed (>twofold change, adjusted *P* value <0.05) between young and old stromal cells of the indicated populations. **b** Heatmap of Metacore Process Networks enriched in three or more aged stromal cell populations at *P* < 0.005. **c** Quantification of myelo-erythroid progenitors from old mice treated with control IgG (IgG; *N* = 9) and neutralising anti-IL-6 antibody (αIL-6; *N* = 10). Data are from five independent experiments. Values are mean ± s.d. *P < 0.05; **P < 0.01; ***P < 0.001 (two-tailed unpaired Student's *t* test). Exact *P* values: MkP: 0.24; preGM: 0.24; GMP: 0.03; preMegE: 0.02; preCFU-E: 0.82; CFU-E: 0.85. **d** In vitro colony-forming potential of LK (defined as Lin⁻Sca-1⁻c-Kit⁺ cells) isolated from BM of old mice after injection of IgG isotype control (*N* = 4) or αIL-6 Ab (*N* = 4) assayed under E condition (M3436; 500 cells). Data are from two different experiments. Values show mean ± s.e.m. ***P < 0.001 (two-tailed unpaired Student's *t* test). Exact *P* value: 0.000000005. **e** In vitro colony-forming potential of LK (defined as Lin⁻Sca-1⁻c-Kit⁺ cells) isolated from BM of old mice after injection of IgG isotype control (*N* = 4) or αIL-6 Ab (*N* = 4) assayed under GM condition (M3534; 500 cells). Data are from two different experiments. Values show mean ± s.e.m. ***P < 0.001 (two-tailed unpaired Student's *t* test). Exact *P* value: 0.0005. **f** In vitro colony-forming potential of preCFU-E cells isolated from BM of old mice after injection of IgG isotype control (*N* = 4) or αIL-6 Ab (*N* = 4) assayed under E condition (M3436; 500 cells). Data are from two different experiments. Values show mean ± s.e.m. ***P < 0.001 (two-tailed unpaired Student's *t* test). Exact *P* value = 0.0005. **g** GSEA analysis comparing expression of preCFU-E-specific genes in preCFU-Es isolated from young and old IgG-treated mice. The normalised enrichment score (NES) and *P* value are shown. **h** GSEA analysis comparing expression of preCFU-E-specific genes in preCFU-Es isolated from young and old anti-IL-6-treated CFU-Es. The normalised enrichment score (NES) and *P* value are shown. Source data are provided as a source data file.

ageing. In particular, we find that TGFβ signalling is critical to the maintenance of age-associated HSC platelet-myeloid lineage bias, and that its inhibition improves lymphoid output from HSCs. In young mice myeloid and erythroid-lineage output from fate-restricted HSCs are correlated[14]. However, during ageing erythropoiesis decreases while myeloid output is increased. We here find that increased TGFβ and IL-6 signalling impair the generation and function of erythroid-lineage progenitors, respectively, providing a mechanism for the distinct effects of ageing on erythroid and myeloid lineage output. Together, these findings identify druggable targets with the potential to improve aged hematopoiesis and in particular show that targeting multiple signalling systems may be required to normalise aged erythropoiesis.

## Methods

**Animals**. All experimental procedures and mouse breeding and maintenance were in accordance with UK Home Office regulations. All experiments were approved by the Oxford University Clinical Medicine Ethical Review Committee. All mice used were from the C57Bl6/J background. For transplantations recipients and competitor cells were of the CD45.1 allotype. Donor mice of the CD45.2 allotype and carried *Vwf*-tdTomato[14] and *Gata1*-EGFP[47] transgenes. Young mice were 2–3 months old and old mice were 24–25 months old. Only female mice were used for experiments. When multiple experimental groups were analysed, mice were allocated so that each group was evenly matched for age range. No statistical methods were used to predetermine the experimental sample size.

**HSC transplantation**. Fifty phenotypically defined LT-HSCs (LSKCD48-CD150⁺ CD34⁻) were isolated from young (2–3 months) or old (24–25 months) *Vwf*-TdTomato/*Gata1*-eGFP (CD45.2) double reporter mice and intravenously injected along with 2 × 10⁵ young wild-type unfractionated C57BL/6 bone marrow cells (CD45.1) into lethally irradiated CD45.1 young recipients (2 × 500 rad). Peripheral blood reconstitution analysis was performed at 16 weeks after transplantation.

**In vivo inhibitor studies**. Aged mice were injected three times intraperitoneally (i.p.) with 100 μg of mouse IL-6 neutralising antibody or Rat IgG2a isotype control (InvivoGen) dissolved in 1 mL of sterile saline solution at 1-week intervals and the mice were culled for analysis 6 days from the last injection. For SB-525334 treatment, aged mice were treated for 13 consecutive days, once daily, with either 30 mg/kg SB-525334 dissolved in corn oil or vehicle (corn oil alone) via oral gavage. The mice were culled for analysis the day following the last gavage.

**Hydrodynamic injection**. Krebs–Ringer modified buffer (pH 7.4) with 0.05% BSA was used as final working buffer for all hydrodynamic injections as previously described[33,48]. The injection volume was ~8–12% of the mouse body weight[49]. After being generally anaesthetised, mice were tail vein-injected in the working buffer with 1 μg of a pCMV6-Entry vector containing IL-6 cDNA (OriGene, cat no. MR227281) or an empty pCMV-entry vector (OriGene, cat no. PS100001). The injected mice were incubated for 7 days before being analysed. Blood samples were collected into eppendorf tubes and left at room temperature to clot for 2 h. The samples were then centrifuged (20 min, 2000 × g, 4 C) and the supernatants were collected and stored at −80 °C immediately. IL-6 serum levels were determined using the IL-6 ELISA Kit (Biolegend, cat no. 431301).

**TGFβ1 ELISA**. Femurs and tibias from mice were flushed with 200 μl of PBS; the flushed BM suspension was centrifuged (5 min, 500 × g, 4 °C) and the BM supernatant was collected. TGF-B1 bone marrow supernatant levels were determined using the Quantikine ELISA Kit (R&D systems, MB100B) according to the manufacturer's instructions.

**Peripheral blood parameters measurement**. Peripheral blood from young and old mice was collected into lithium heparin-coated microvettes (Sarstedt) and diluted five times in PBS. White blood cells (WBC), Red blood cells (RBC) and platelet counts were measured using a Sysmex KX-21N Automated Haematology Analyser (three replicates/sample). Hb and HCT values were obtained in the same way (three replicates/sample).

**Flow cytometry and cell sorting**. Details of mouse antibodies and viability dyes used for each staining panel are shown in Supplementary Data 3. All antibodies were used at optimal working concentrations, determined by titration. Hematopoietic stem and progenitor cells, myelo-erythroid progenitor cells, peripheral blood WBC, RBC and PLT were analysed as previously described[14,47]. The optimal gating strategy was chosen using a combination of fluorescence minus one controls and populations known to be negative/positive for the antigen. Cell acquisition and analysis were performed using LSRII or LSR Fortessa X-20 cytometer (BD Biosciences) using BD FACSDiva™ software (BD Biosciences). Phenotypically defined LT-HSCs, LK, LSK and preCFU-E were sorted using a BD FACSAria II cell sorter (BD Biosciences). Analysis was performed using Flowjo software version 10.3 (Flowjo LLC, OR, USA).

**HSC cell cycle analysis**. For HSC cell cycle analysis, BM cells were fixed and permeabilized using BD cytofix/cytoperm fixation and permeabilization solution (BD Biosciences; BD 554714) following initial Fc-block incubation. Fixed cells were then stained with Ki-67 PE (BD Biosciences; BD 556027) overnight. The day after, cells were washed and stained with 4′,6-diamidino-2-phenylindole (0.5 mg/mL) (ThermoFisher Scientific, Waltham, MA, USA) for 1 h before acquisition. Cell acquisition and analysis were performed using LSRII or LSR Fortessa X-20 cytometer (BD Biosciences) using BD FACSDiva™ software (BD Biosciences). Analysis was performed using Flowjo software version 10.3 (Flowjo LLC, OR, USA).

**Early B precursor isolation**. BM cells were stained with biotinylated antibody B220 Biotin (eBioscience, 13–0452–82) following initial Fc-block incubation. After incubation, cells were washed twice with PBS/5% FCS, centrifuged and incubated with Anti-Biotin MicroBeads (Miltenyi Biotec, 130–090–485). Magnetic purification was performed according to the manufacturer's instructions. B220+ cells were counted and stained with SAv-Brilliant Violet 605™ (BioLegend, 405229) and anti-Mouse IgM PE-Cyanine7 Antibody (Life Technologies, 25–5790–81). After wash, cells were stained with 7-Aminoactinomycin D (7AAD; Biotium 40037) and phenotypically defined early B precursors (7AAD⁻B220⁺IgM⁻) sorted using a BD FACSAria II cell sorter (BD Biosciences). Analysis was performed using Flowjo software version 10.3 (Flowjo LLC, OR, USA).

**B-cell progenitor proliferation assay**. Proliferation was measured using the CellTiter 96® AQueous One Solution Cell Proliferation Assay (MTS) (Promega, G3582) following the manufacturer's instructions. In detail, phenotypically defined early B precursors (100,000 cells/well) were sorted in 96-well flat-bottom plates in 0.1 ml Opti-Mem supplemented with 10% FBS, penicillin–streptomycin (100 U/ml), L-glutamine (2 mM) and 2-mercaptoethanol (50 μM) using a BD FACSAria II cell sorter (BD Biosciences). mIL-7 (R&D Systems) was added at different concentrations (0.1–1–10–100 ng/ml). After 48 h, 20 μl of CellTiter 96® AQueous One

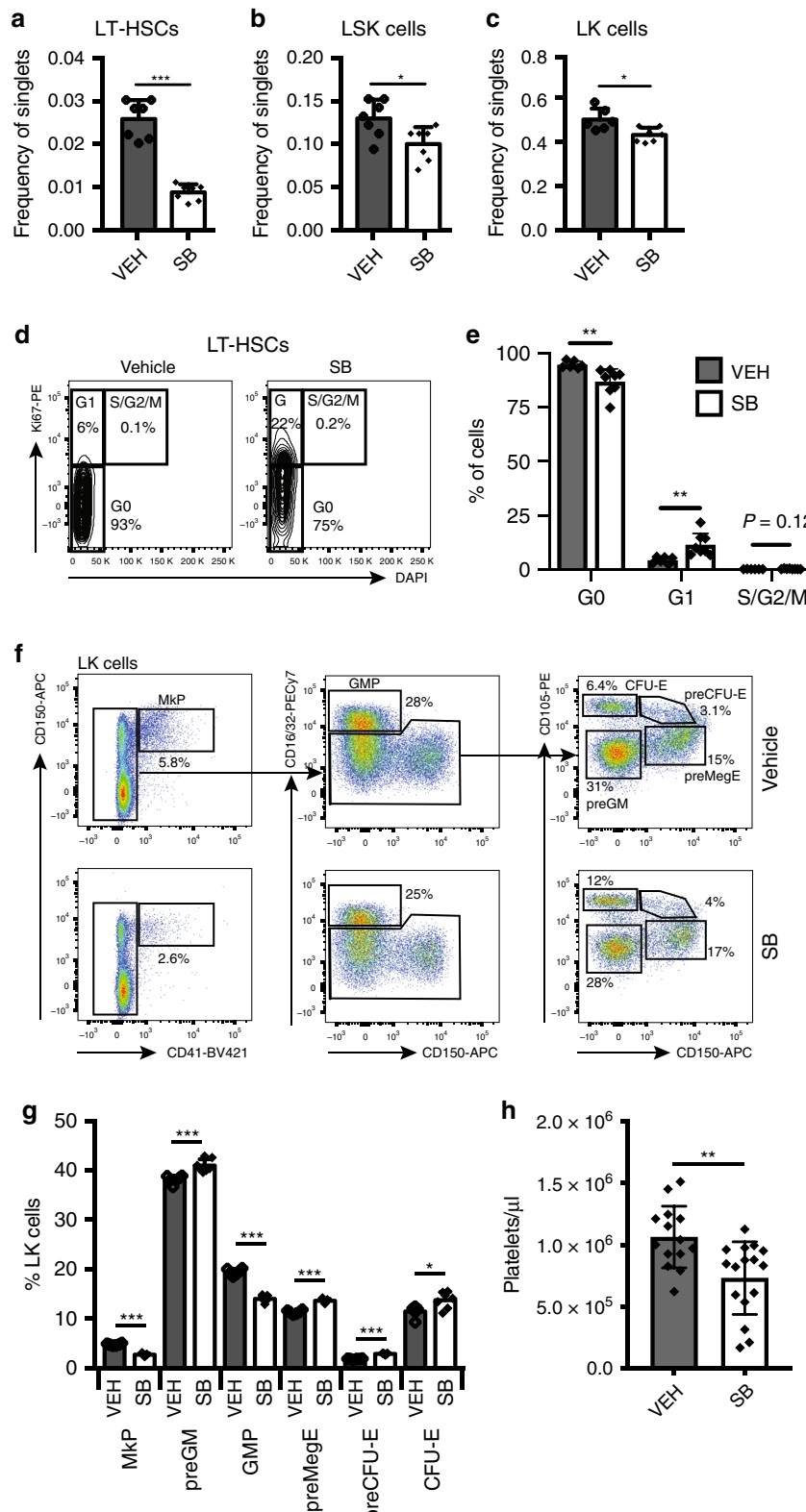

Solution Reagent was added into each well of the 96-well assay plate containing the samples, plates were incubated at 37 °C for 4 h in a humidified, 5% $CO_2$ atmosphere and the absorbance at 490 nm was recorded.

**In vitro erythroid/myeloid colony-forming assay**. Media used for erythroid/myeloid colony-forming assays were from STEMCELL Technologies and were used according to the manufacturer's instructions. Overall, 500 cells were seeded per culture in M3436 (supplemented with SCF 100 ng/ml) and M3534, and the

colonies generated in methylcellulose were scored by morphology as CFU-E and CFU-GM, respectively after 7 days. Cells were plated in triplicate for each assay.

**In vitro lymphoid colony-forming assay**. Media used for lymphoid colony-forming assay was from STEMCELL Technologies and was used according to the manufacturer's instructions. In total, 25,000 phenotypically defined early B precursors ($7AAD^-B220^+IgM^-$) were seeded per culture in M3630 (supplemented with SCF 20 ng/ml) and the colonies generated in methylcellulose were scored by morphology after 7 days. Cells were plated in triplicate.

**Fig. 4 Effect of TGFβR1 inhibition on aged hematopoiesis.** Bar graphs showing the number of LT-HSCs (**a**), LSK (**b**) and LK (**c**) cells in bone marrow of old mice treated with vehicle (VEH) or SB (LT-HSCs and LSK: $N = 7$/condition; LK: VEH $N = 6$, SB $N = 7$) as frequency of live single cells. Data are from three independent experiments. Values show mean ± s.e.m. *$P < 0.05$; ***$P < 0.001$ (two-tailed unpaired Student's $t$ test). Exact $P$ values: LT-HSCs: 0.00000072; LSK: 0.0200; LK: 0.011. **d** Representative flow cytometry plots of LT-HSC cell cycle distribution from vehicle and SB-treated old mice. Gating for G0, G1 and S/G2/M cell cycle phases is shown, as are the $P$ values for comparison between the two conditions. **e** Quantification of LT-HSCs in each phase of the cell cycle from vehicle (VEH) and SB-treated old mice (VEH: $N = 6$; SB: $N = 8$; three independent experiments). Values are mean ± s.d. **$P < 0.01$ (two-tailed unpaired Student's $t$ test). Exact $P$ values: G0: 0.0092; G1: 0.0072; S/G2/M: 0.1206. **f** Representative flow cytometry plots of myelo-erythroid progenitors from vehicle (top panels) and SB-treated old mice (bottom panels). **g** Quantification of myelo-erythroid progenitors from vehicle (VEH) and SB-treated old mice (VEH: $N = 6$; SB: $N = 7$; three independent experiments). Values are mean ± s.e.m. *$P < 0.05$; ***$P < 0.001$ (two-tailed unpaired Student's $t$ test). Exact $P$ values: MkP: 0.000000005; preGM: 0.0007; GMP: 0.00000001; preMegE: 0.00000024; preCFU-E: 0.0000033; CFU-E: 0.0148. **h** PB platelet count in vehicle (VEH) and SB-treated old mice (VEH: $N = 14$, SB: $N = 16$). Data are from seven independent experiments. Values are mean ± s.e.m. **$P < 0.01$; (two-tailed unpaired Student's $t$ test). Exact $P$ value: 0.0026. Source data are provided as a source data file.

**In vitro B-cell lineage potential assay**. B-cell lineage potential was evaluated in vitro in OP9 stromal co-cultures. OP9 stromal cells were maintained in adherent cultures in 75 cm$^2$ flasks (Corning) in Opti-MEM with Glutamax medium (Gibco), supplemented with 10% FCS, 1% Penicillin/Streptomycin (final concentration 100 U/ml/0.1 mg/ml) and 1% 2-mercaptoethanol (final concentration 10$^{-4}$M). One day before sorting the cells of interest for the assay, OP9 stromal cells were trypsinized (Trypsin EDTA, PAA Laboratories) and monolayers were obtained seeding $2 \times 10^3$ cells per well in flat-bottom 96-well-plates (Corning Costar). LSK cells were FACS sorted from bone marrow of young and old mice and plated at a density of 10 cells per well into the wells previously prepared containing monolayers (80% confluence) of OP9 stromal cells. Cultures were maintained for 3 weeks in culture medium supplemented with mSCF (5 ng/ml), hFL (5 ng/ml) and hIL-7 (2 ng/ml) and B-cell potential assessed by flow cytometry using LSR Fortessa X-20 cytometer (BD Biosciences). Details of mouse antibodies and viability dyes used for the staining panel are shown in Supplementary Data 3.

**Peripheral blood reconstitution analysis**. Peripheral blood reconstitution in transplant recipients was analysed by flow cytometry at 16 weeks after transplantation for all mice. Peripheral blood from tail vein was collected into lithium heparin-coated microvettes (Sarstedt) and processed as previously described[14]. Briefly, a small aliquot of unfractionated peripheral blood was used for analysis of erythroid cells, separately from platelet solution. Platelets were separated by centrifugation of peripheral blood samples at $100 \times g$ for 10 min at room temperature. For leucocyte separation, samples were incubated 1:1 with 2% w/v Dextran (Sigma-Aldrich), for 20–30 min at 37 °C. Left erythrocytes were lysed from the leucocyte preparation through incubation with ammonium chloride solution (STEMCELL Technologies) for 2 min at room temperature, and leukocyte samples were pre-incubated with Fc-block before staining. Anti-mouse antibody staining for reconstitution analysis by flow cytometry was carried out for 15–20 min at 4 °C PBS/5% FCS buffer and samples were analysed using LSRII or LSR Fortessa X-20 flow cytometers (BD Biosciences). Details of mouse antibodies used for the staining panel are shown in Supplementary Data 3.

**Central bone marrow stromal cells isolation**. Adapted versions of the protocols described[50–52] were used to isolate central bone marrow (sinusoidal) and BL (endosteal) niche populations. To isolate central bone marrow stromal cells, dissected bones were crushed twice in a pestle and mortar containing 5 mL PBS/5% FCS buffer, aspirated and passed through a 50-μm mesh filter into a fresh 15 mL Falcon. After crushing, remaining bone fragments were kept for isolating BL niche cells. The filtered BM solution was counted using a Sysmex KX-21N Automated Haematology Analyser, centrifuged (5 min, $500 \times g$, 4 °C) and resuspended in 5 mL Dulbecco modified eagle medium (DMEM) containing 10% FCS, 200 U/mL DNase I (STEMCELL Technology) and 10 mg of collagenase type IV (Worthington Biochemical) for 20 min at 37 °C on a rotating plate. After incubation, cells were washed with PBS/5% FCS and centrifuged (5 min, $500 \times g$, 4 °C). The pellet was resuspended in 5 mL of ammonium chloride solution (STEMCELL Technology) for 10 min at room temperature. The lysis reaction was neutralised using 5 mL of PBS/5% FCS, centrifuged (5 min, $500 \times g$, 4 °C), resuspended in 5 mL PBS/5% FCS and counted. Depletion of the CD45$^+$ cell fraction was carried out in order to improve the central bone marrow niche cell recovery. Cells were resuspended in 90 μL PBS/5% FCS and 10 μL anti-CD45 microbeads (Miltenyi, cat no. 130–052–301) per $10 \times 10^6$ cells. The solution was gently agitated every 5 min and incubated for a total of 20 min at 4 °C. During this time, LS columns (Miltenyi) were set up on a magnetic stand and equilibrated with 10 mL PBS/5% FCS. The cell solution was then washed with PBS/5% FCS, centrifuged (5 min, $500 \times g$, 4 °C) and resuspended in 2 mL PBS/5% FCS. Once completely clear of any remaining solution, the LS column was loaded with the cell suspension and collected in a fresh 15 mL Falcon tube. Cells were centrifuged (5 min, $500 \times g$, 4 °C), counted and stained for FACS using the antibodies outlined in Supplementary Data 3.

**BL stromal cells isolation**. Remaining bone fragments left after central bone marrow stromal cells isolation were minced into smaller fragments using scissors, resuspended in 5 mL of PBS/5% FCS and centrifuged (5 min, $500 \times g$, 4 °C). Bone fragments were resuspended in 5 mL DMEM containing 10% FCS, 200 U/mL DNase I (STEMCELL Technology) and 15 mg of collagenase type I (Worthington Biochemical) for 45 min at 37 °C on a rotating plate. Digested bone fragments were allowed to settle and the top layer was aspirated, passed through a 70-μm mesh filter into a 50 mL Falcon tube and sored at 4 °C. Collagenase treatment was repeated once more on the remaining bone material to ensure maximum niche cell recovery. The resulting cell solution was washed with 10 mL PBS/5% FCS and centrifuged (5 min, $500 \times g$, 4 °C). Cells were then stained for FACS using the antibodies outlined in Supplementary Data 3.

**Generation of cDNA libraries using Smart-seq2 protocol**. An adapted version of the low-cell number SMART-seq2 protocol[53,54] was used for processing bulk RNA-seq samples. For all bulk RNA-seq experiments, 100 cells/biological replicate were isolated by FACS directly into thin-walled PCR tubes containing 4 μl of lysis buffer containing: 0.2% Triton X-100 (Sigma-Aldrich), 2.5 μM OligodT (Biomers), 2.5 mM dNTPs (Clontech) and RNase Inhibitor 4U (Clontech). Immediately after sorting, samples were vortexed, centrifuged ($700 \times g$, 10 s, room temperature), and placed at −80 °C for a maximum of 4 weeks before being further processed. For the reverse transcriptase, 6 μl of the following mix was added: 2 μl Superscript II first strand buffer, 0.5 μl DTT (100 mM), 2 μl Betaine (5 M), 0.1 μl MgCl$_2$ (1 M), 0.25 μl RNase Inhibitor (40 U/μl), 0.1 μl TSO (100 μM), 0.25 μl Superscript II RT (200 U/μl) and 0.4 μl water. After reverse transcriptase, 15 μl was added containing 12.5 μl Kapa HiFi HS ReadyMix (2×) and 0.125 μl ISPCR primers (10 μM). The five thermal conditions for RT and pre-amplification were according to the original Smart-seq2 protocol[53,54]. The number of cycles used for PCR amplification was 18. After PCR amplification, cDNA libraries were purified using Ampure XP magnetic beads according to the manufacturer's instructions. After purification, the libraries were resuspended in 17.5 μl of buffer EB (Qiagen) and stored at −20 °C. Quality and concentration of the cDNA libraries generated was assessed using High-Sensitivity Bioanalyzer (Agilent).

**Illumina library preparation and sequencing**. In total, 0.7 ng of cDNA was tagmented using the Nextera XT DNA Sample Preparation kit (Illumina) according to the manufacturer's instructions, except that one-fourth of the volumes indicated were used. Purification of the product was done with a 1:1 ratio of AMPure XP beads, with a final elution in 17.5 μl in resuspension buffer provided from the Nextera kit. Samples were loaded on a High-Sensitivity DNA chip (Agilent Technologies) to check the size and quality of the indexed library, and the concentration was measured with the Qubit High-Sensitivity DNA kit (Invitrogen). Libraries were pooled to a final concentration of 4 nM and were sequenced with Illumina NextSeq 500 (76 bp single-end) or Illumina HiSeq4000 (50 bp single-end) after preparation according to manufacturer's instructions.

**Data analysis**. Bulk RNA-seq samples were sequenced on Illumina NextSeq 500 (76 bp, single-end) or Illumina HiSeq4000 (50 bp, single-end) platforms. FASTQ files were inspected using FastQC (v0.10.1) followed by Nextera adaptor sequence removal using TrimGalore! (v0.4.1; http://www.bioinformatics.babraham.ac.uk/projects/trim_galore) with a stringency setting of 3. Reads were aligned to the mm9 transcriptome using STAR (v2.4.2a)[55]. Unique reads were counted using featureCounts (v1.4.5-p1)[56] and the UCSC mm9 annotation file. All output files were quality assessed using MultiQC (v0.7)[57]. For all bulk RNA-seq samples, read counts were imported into R Studio and a local fit method was used for differential gene expression analysis. Counts were normalised using the rlog Transformation function in DESeq2 (v1.14.1)[58] with the blind setting set to true. This function log transforms data and normalises gene expression with respect to library size. Reads per Kilobase of transcript per Million mapped reads (RPKM) values were generated using the EdgeR::rpkm function (v2.16.5[59]). Metacore v6.35 (Thomson Reuters, www.portal.genego.com) was used to determine enriched process networks.

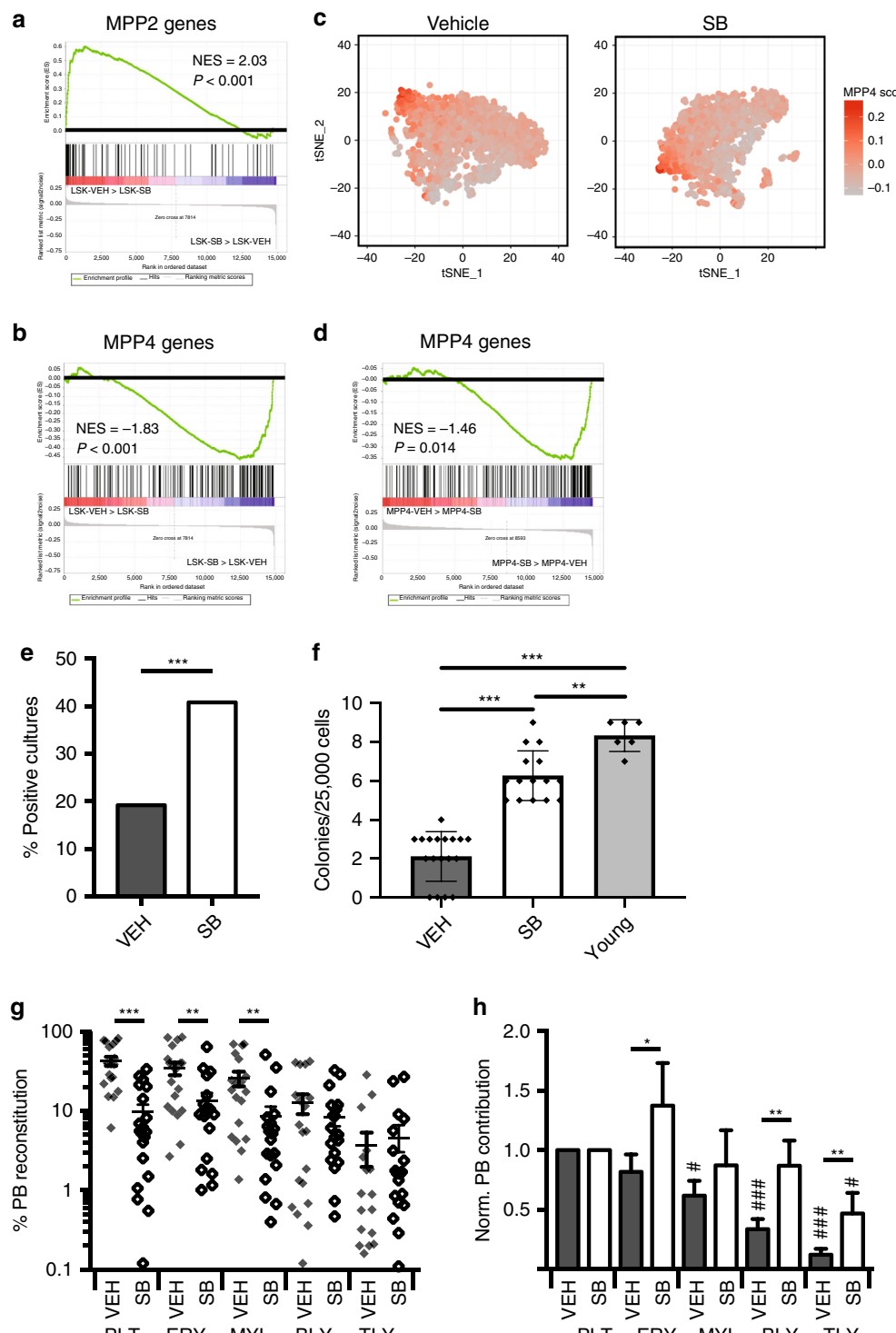

Ranked DEGs (adjusted $P \leq 0.05$) were used as input for Metacore, which models functionally enriched ontology based on a hypergeometric distribution model. After calculating a $P$ value for all terms within a given ontology, each term is then tested as a separate hypothesis which results in the generation of a $q$ value. This subsequently provides an estimate of false discovery rate (FDR). All Metacore output displayed in this study have both a $P \leq 0.05$ and FDR $\leq 0.05$. GSEA (v3.0[60]) was performed using the javaGSEA application. RPKM values were imported as input data. Gene sets for GSEA were taken from MSigDB (IL-6 gene set M14344) and Mancini et al.[61].

**Single cell sequencing library preparation**. LSK cells were FACS isolated from aged (24–25 months old) mice treated with vehicle or SB-525334 and processed using the 10× Genomics droplet-based sampler. Library preparation was

performed following the Chromium Single Cell 3′ Reagents Kit v2 users guide (CG00052, Rev C). Quantification of cDNA was done using the Qubit dsDNA HS kit (Invitrogen) and a Qubit fluorometer (Invitrogen). Fragment sizes were determined using an Agilent high-sensitivity DNA chip (Agilent Technologies) and an Agilent 2100 Bioanalyzer (Agilent Technologies). After pooling, library quantification was determined using the KAPA qPCR platform (Kapa Biosystems).

**10× Chromium data analysis**. Alignment, quality checking, filtering and analysis of UMI-based scRNA-seq datasets was achieved using the following steps. Demultiplexing and alignment: Sequencing of UMI-based cDNA libraries from young and aged unfractionated LSK was carried out using the Illumina HiSeq4000 platform (75 bp paired-end). Sample demultiplexing was completed by the Wellcome Trust Centre for Human Genetics, University of Oxford, and FASTQ files

**Fig. 5 TGFβR1 inhibition rebalances aged HSC lineage output. a** GSEA analysis of MPP2-specific genes in LSK cells from vehicle (VEH) and SB-treated old mice. Normalised enrichment score (NES) and P value are shown ($P < 0.001$). **b** GSEA analysis of MPP4-specific genes in LSK cells isolated from vehicle (VEH) and SB-treated old mice. Normalised enrichment score (NES) and P value are shown ($P < 0.001$). **c** tSNE plot of vehicle and SB-treated single LSK cell transcriptomes. **d** GSEA analysis of MPP4-specific genes in computationally defined MPP4 cells isolated from vehicle and SB-treated old mice. The normalised enrichment score (NES) and P value are shown ($P = 0.014$). **e** Fraction of LSK cultures from vehicle and SB-treated old mice (10 cells plated/culture) positive for B-lymphocytes after 3 weeks of culture. $N = 192$ cultures/condition, two independent experiments. ***$P < 0.001$ (two-sided $\chi^2$ test). Exact P value: 0.0004. **f** In vitro colony-forming potential of B precursor cells isolated young mice ($N = 6$), vehicle ($N = 18$) and SB-treated ($N = 15$) old mice after plating 25.000 cells in M3630 medium. Data from two different experiments. Values show mean ± s.e.m. **$P < 0.01$; ***$P < 0.001$ (two-tailed unpaired Student's t test). Exact P values: VEH-SB: 0.0000000002; SB-Young: 0.0017; VEH-Young: 0.0000000002. **g** PB lineage contribution of LT-HSCs from vehicle (VEH) and SB-treated (SB) old mice in transplant recipients (VEH: $N = 19$; SB: $N = 20$; three independent experiments). Mean values ± s.e.m are shown. **$P < 0.01$; ***$P < 0.001$ (two-sided Mann–Whitney U-test). Exact P values: PLT: 0.000002; ERY: 0.002; MYL: 0.002; BLY: 0.66; TLY: 0.41. **h** Donor HSC PB contribution of LT-HSCs isolated from vehicle (VEH) and SB-treated (SB) old mice in transplant recipients (VEH: $N = 19$; SB: $N = 20$; three independent experiments) from **g** normalised to platelet output. Mean values ± s.e.m are shown. *$P < 0.05$; **$P < 0.01$ (two-sided Mann–Whitney U-test). Exact P values: PLT: not applicable; ERY: 0.01; MYL: 0.38; BLY: 0.006; TLY: 0.002. P values for the comparison of each lineage output to platelet output for the same condition are also shown: #$P < 0.05$; ###$P < 0.001$ (two-sided Mann–Whitney U-test). Exact P values, VEH condition: ERY: 0.36; MYL: 0.02; BLY: 0.0001; TLY: 0.000002. Exact P values, SB condition: ERY: 0.31; MYL: 0.55; BLY: 0.74; TLY: 0.03. Source data are provided as a source data file.

were aligned to the mm10 reference genome using CellRanger (v2.0.1; 10× Genomics). Biological replicates from the same conditions were aggregated using the CellRanger::Aggregate function. Filtering, merging and normalising data: Filtered gene-cell matrices of aggregated samples were imported into R Studio using the Seurat::Read10× function (v2.3.0[62]). tSNE was run to visualise the gene expression of single cells in two-dimensional space. Differential gene expression analysis was carried out using a negative binomial statistical test[63]. DEGs with an adjusted $P < 0.05$ and LogFC > 0.25 were returned and used for subsequent analyses. GSEA[60] was used to determine enriched functions. DEGs were ranked by log2 fold change (Log2FC) and imported into the GSEAPreranked tool. The MPP2 and MPP4 gene sets were generated from published expression data, comparing published HSC, MPP2, MPP3 and MPP4 gene expression. Genes considered specific for a given population were those significantly upregulated ($P_{adj} < 0.05$) when compared to the other three populations (Supplementary Data 4)[40]. MPP4 scores for single cells were generated using the Seurat AddModuleScore function and the MPP4 gene set defined above, with MPP4 cells defined as those scoring above the cutoff that identifies the same proportion of MPP4 cells in young mice as flow cytometry[40].

**Statistical analysis**. Statistical comparisons of continuous data with assumed normal distribution (mean percentages) were performed with parametric unpaired two-tailed Student's t test. Two-sided Mann–Whitney U-test was used for data with non-normal distribution. Statistical comparisons of categorical data (frequencies) were performed with nonparametric two-tailed $\chi^2$-test. Student's t test, Mann–Whitney U-test and $\chi^2$-test were performed with GraphPad Prism software or in R using the appropriate R function. Only statistically significant differences ($P < 0.05$) are indicated in the figures.

**Reporting summary**. Further information on research design is available in the Nature Research Reporting Summary linked to this article.

## Data availability

The RNA sequencing data generated have been deposited in the GEO database under the accession code GSE130899. Further details are available from the corresponding authors upon reasonable request. Source Data are provided with this paper.

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

## Acknowledgements

We thank Dr. Joana Carrelha and Prof. Sten Eirik Jacobsen for providing GE/VT mice and for helpful discussions, Dr. Tiago C. Luis for sharing protocols for niche cell isolation, and the WIMM flow cytometry core facility and the Biomedical Services at University of Oxford for technical support. This work was supported by a BBSRC Project Grant (BB/M024350/1) and a MRC Unit Grant (MC_UU_12009/7) to C.N., and a Bloodwise Gordon Piller PhD studentship to A.T. WIMM FACS Core Facility is supported by the MRC HIU, MRC MHU (MC_UU_12009), NIHR Oxford BRC and the John Fell Fund (131/030 and 101/517), the EPA fund (CF182 and CF170) and by WIMM Strategic Alliance awards (G0902418 and MC_UU_12025).

## Author contributions

S.V., A.T., Y.M., X.R., R.D., H.S. and C.G. performed the experiments. A.T., Y.M. and C.N. performed RNA sequencing data analysis. C.N. and S.V. conceived the experiments and wrote the paper.

## Competing interests

The authors declare no competing interests.
