## [Peer Review File · Nature Communications]

Reviewers' comments:

Reviewer #1 (Remarks to the Author):

In this manuscript, Valletta et al. analyzed the stromal cells of the bone marrow to address the decline of erythropoiesis and lymphopoiesis upon ageing. They found upregulation of IL-6 and TGF β signaling induced gene expression in BM stromal cells of aged mice, and could subsequently demonstrate that both cytokines play an important, but distinct role in this process. The authors demonstrate HSC-intrinsic and extrinsic mechanisms that drive hematopoietic decline during ageing, and associate the BM stroma in this process. The amount of work presented is substantial, and the fact that the involvement of not one, but two cytokines is investigated, provides extra value to this manuscript. This study is important for the field, as it helps us to understand how vital physiological processes deteriorate during ageing, and what could be done to counteract this.

The manuscript is very well written, and the experiments are of a high technical level and well-performed. Data is generally well displayed and conclusions are in line with the findings. Although most of the findings and claims are novel, it should be noted that the contribution of IL-6 to erythroid decline during ageing has been shown before in several studies (some of them also referenced by the authors; Refs 21-24).

Eventhough not all of the observations are experimentally addressed and still remain unclear (eg. what is the contribution of the other inflammatory networks, beyond IL-6 and TGF β , that are also observed in BM stromal cells of aged mice?), it is obvious that gene expression analysis evokes new questions for future studies. Hence, I restrict my feedback to the questions that the authors experimentally addressed and the conclusions they drew from their experiments.

Major comments:

A. Based on the stromal cell analysis, the authors hypothesized that increased IL-6 signaling could underly (part of) the ageing phenotype. Indeed, based on blocking experiments with anti-IL-6 antibodies (Figure 3), the authors conclude that IL-6 is an important driver of the age-mediated decline in erythropoiesis, which agrees with several other studies. However, it is not clear whether the authors now conclude that IL-6 affects erythropoiesis through the BM stroma or that it acts directly on the erythroid progenitor cells? It could well be that the observed IL-6 mediated changes in BM stroma are an epiphenomenon and not causally related to the erythroid decline, but this should be made clear, given the important focus in the manuscript on the stromal cells in the BM.

B. The authors argue that TGF β acts independently of IL-6, and plays a different role in the ageing of HSCs, LSK and myelo-erythroid progenitor compartments; here, it is made likely that the stroma itself is actively contributing, as the authors found more TGF- β 1 in ageing compared to young stroma. However, it has been well established that TGF β is a major determinant of HSC quiescence, which is neither referred to nor experimentally addressed here. It is conceivable that the effects observed with the TGF β R-inhibitor in Figures 4 and 5 are simply due to a loss of HSC quiescence, which subsequently resulted in enhanced/increased differentiation of the HSCs and downstream progenitors. It is important that the authors adequately address this possibility and, if necessary, adjust their conclusions accordingly.

Minor comments:

C. The authors show a lot of GSEA graphs, but it is unclear for the reader which genes are part of the indicated enrichment. For instance, in Figure 2h there is an increase in so-called "preGM-associated genes" in CFU-Es from old mice, but the identity of these genes is not revealed and it is also not clear which genes within these gene-sets are indeed higher or lower. It would be highly valuable if the GSEA analyses that are shown throughout this manuscript would be expanded by including the related genes in a supplemental figure or table.

D. In figure 2, the authors describe that purified MPP2, preCFU-Es and CD71+ CFU-Es from aged mice

all have decreased erythroid differentiation (Figure 2f). The prediction is that they all show a gain in myeloid differentiation, but this is only shown for the preCFU-E population in Figure 2i. It would be worth to also show the myeloid potential of the MPP2 population, as this would reveal how far up in the hematopoietic hierarchy the switch from erythroid to myeloid occurs.

E. In Figure 3g&h, the authors show in a GSEA that the loss of preCFU-E genes upon ageing is prevented by depletion of IL-6. It is important to also show whether the enrichment of preGM genes upon ageing (Fig 2h) is also lost by the IL-6 depletion. This would denote whether IL-6 truly affects a shift in the balance between erythroid and myeloid commitment, or only affects erythroid differentiation.

F. At the bottom of page 5, the authors refer to Suppl Fig 3e, but this does not exist and should be Suppl Fig 4a.

Reviewer #2 (Remarks to the Author):

In the manuscript, "Using bone marrow stromal cells as micro-environmental sensors identifies IL-6 and TGF β signaling as regulators of declining erythropoiesis and lymphopoiesis during hematopoietic ageing," Valletta et al. utilized bone marrow (BM) stromal cells to determine consistent differences in cellular signaling across multiple cell types between young and aged BM and find enhanced IL-6 and TGF beta gene programs in aged BM. Next, hematopoietic stem cells (HSC) were assessed functionally by transplantation to determine if age related defects in erythropoiesis and lymphopoiesis are cell-autonomous, and the author observed small non-significant changes in the intrinsic ability for aged HSC to initiate erythropoiesis while declining lymphopoiesis was a cell-autonomous effect. Taken together these data indicated both cell-autonomous and non-cell-autonomous effects on the ability of aged HSC to produce normal lineage output.

Overall, the work shows some important findings relating to cell-extrinsic signaling networks in aging impacting platelet and erythroid output, which are often ignored in favor of myeloid vs. lymphoid lineage alterations only.

Major concerns:

Inflammatory gene signatures in aging HSC are well described in prior work (Chambers, Plos Biology 2007) Here, RNA-seq analysis of stromal cells identified a similarly inflammatory signature that impacts hematopoiesis. Thus, the value of analyzing gene expression in the stroma or calling them 'sensors' is not clear, given gene expression analysis of HSC/HSPC would yield similar results. Value would be added to this analysis if the impact of the inflammatory signature on HSC maintenance function of the stromal cells themselves is addressed. Indeed a larger issue with the work as a whole is that a preliminary level of mechanistic insight with respect to the impact of TGF-beta and/or IL-6 on aged hematopoietic function is provided.

Increased levels of IL-6/TGF (as opposed to increased signaling activity in these pathways due to cell-intrinsic changes) in the BM are not shown. Indeed other previous analysis of aged BM (Ergen et al, Blood 2012) show decreased IL-6 levels in BM fluid by direct interrogation, whereas the study cited by the authors (Mei et al, 2018) only shows protein levels in serum and not BM (indirect measurement of IL-6 was performed by qRT-PCR in this study). Echoing the point above relating to mechanism, it will be important for the authors to distinguish between the alternative hypotheses of increased levels of these factors in situ in the BM vs. increased signaling activity due to intrinsic deregulations.

Reviewer #3 (Remarks to the Author):

In this paper from Valetta et al., the authors attempt to find an answer to the molecular features underlying ageing hematopoiesis/aging HSC behavior. The studies try to argue for a role for IL6 and Tgfb signaling in controlling hematopoiesis at two levels (erythroid progenitor cells and HSCs, respectively). The interpretations appear for the most part to be supported by the data provided and statistics (where applied) seem adequate.

The authors first describe the general features of ageing hematopoiesis, focusing on decreased erythropoiesis and enhanced platelet (biogenesis?). They validate previous findings and apply a transgenic mouse model that permit tracing of these two lineages (demonstrating that also this transgenic strain exhibits these well-known age-associated features) (Figure 1). In essence, the authors conclude that while there is a general reduction of the contribution of aged HSCs to multilineage hematopoiesis, this is more pronounced for the lymphoid lineages.

Not sure if there is anything wrong with the resolution in my figures, but I cannot really see any cells in the gates connected to Figure 1b (plots to far right) and Figure 1d (CD11b vs Gr1 plot; no myeloid cells?).

Having verified/established these basic features, the authors next investigate myeloerythroid progenitor cells in young and aged mice. They demonstrate an increase in megakaryocyte progenitors and a decrease of early erythroid progenitors (including MPP2s – it could benefit the general reader to explain a bit more why these cells were investigated). Finally, authors conduct some gene expression studies and culture experiments to demonstrate that the myeloid activity of aged preCFU-Es is higher from aged cells. These latter experiments seem to indicate that the myeloid (as opposed to erythroid) activity/cell goes up with age. This raises some concerns as to the way the prospective isolation was made – if f.i. CD55 had been included (positivity) as a marker for candidate preCFU-Es, would the authors obtain the same results?

The authors next go on to use the rather clever approach of using stromal/microenvironmental cells as probes for what might underlie the hematopoietic phenotypes of aging. This lead to the identification of two candidate signaling pathways; IL6 and Tgfb. Further experimentation is focusing on the neutralisation of the two pathways, followed by re-evaluation of hematopoiesis. From these lines of studies, the conclusions are: 1) that IL6 exerts negative effects of erythropoiesis (rather than having direct effects on HSCs) while 2) Tgfb-inhibition appears to restore the lineage (or rather progenitor) output from aged HSCs (including the overproduction of megakaryocyte progenitors; what about platelets?). They provide some molecular evidence (several types of genome-wide RNA expression experiments) to support their conclusions.

Finally, the investigators conduct transplantation experiment using purified HSCs from vehicle or Tgfb-inhibited treated mice. They try to demonstrate that this treatment “normalizes” the output from HSCs with enhanced relative output of non-platelet lineages.

It appears that the overall reconstitution of Tgfb-inhibited cells is dramatically lower than that from vehicle-treated. What is going on here? It appears to this reviewer that Tgfb-inhibition is causing an overall decline in HSC function, that perhaps has some selectivity for HSCs with a “myeloid-bias”/“compromised lymphopoiesis”. But this is very difficult to conclude if the variation in overall reconstitution is very compromised – f.i. low-level chimeras tend to be skewed towards “lymphoid-biased” cells (which is most likely reflecting more on the longevity of mature lymphoid cells).

Some emphasis is on the “rescue” of B lymphopoiesis (Figure 5e) as a consequence of Tgfb-inhibition, but details on whether there is restoration of B lymphopoiesis in the bone marrow is lacking. It would have been very interesting for the authors to provide some information specifically on IL7-signaling, as compromised IL7 signaling in the past has been proposed to be a major mechanism for the reduced B (and/or T) lymphopoiesis with age.

Reviewer #1 (Remarks to the Author):

In this manuscript, Valletta et al. analyzed the stromal cells of the bone marrow to address the decline of erythropoiesis and lymphopoiesis upon ageing. They found upregulation of IL-6 and TGF β signaling induced gene expression in BM stromal cells of aged mice, and could subsequently demonstrate that both cytokines play an important, but distinct role in this process. The authors demonstrate HSC-intrinsic and extrinsic mechanisms that drive hematopoietic decline during ageing, and associate the BM stroma in this process. The amount of work presented is substantial, and the fact that the involvement of not one, but two cytokines is investigated, provides extra value to this manuscript. This study is important for the field, as it helps us to understand how vital physiological processes deteriorate during ageing, and what could be done to counteract this.

The manuscript is very well written, and the experiments are of a high technical level and well-performed. Data is generally well displayed and conclusions are in line with the findings. Although most of the findings and claims are novel, it should be noted that the contribution of IL-6 to erythroid decline during ageing has been shown before in several studies (some of them also referenced by the authors; Refs 21-24). Eventhough not all of the observations are experimentally addressed and still remain unclear (eg. what is the contribution of the other inflammatory networks, beyond IL-6 and TGF β , that are also observed in BM stromal cells of aged mice?), it is obvious that gene expression analysis evokes new questions for future studies. Hence, I restrict my feedback to the questions that the authors experimentally addressed and the conclusions they drew from their experiments.

General comments: We appreciate the reviewer's positive comments. We would, however, like to point out that previous studies on the role of IL-6 in age-related erythroid decline have been correlative – a important novel aspect of the present study is that inhibition of IL-6 is carried out in an aged environment, with a positive effect on erythroid progenitor function, providing direct *in vivo* functional validation of the role of IL-6 in this process.

Major comments:

A. Based on the stromal cell analysis, the authors hypothesized that increased IL-6 signaling could underly (part of) the ageing phenotype. Indeed, based on blocking experiments with anti-IL-6 antibodies (Figure 3), the authors conclude that IL-6 is an important driver of the age-mediated decline in erythropoiesis, which agrees with several other studies. However, it is not clear whether the authors now conclude that IL-6 affects erythropoiesis through the BM stroma or that it acts directly on the erythroid progenitor cells? It could well be that the observed IL-6 mediated changes in BM stroma are an epiphenomenon and not causally related to the erythroid decline, but this should be made clear, given the important focus in the manuscript on the stromal cells in the BM.

The reviewer raises a very relevant point. While we here use the stromal cells as indicators for age-related changes to the microenvironment those changes obviously also have the potential to alter stromal cell function, and thereby indirectly hematopoiesis itself. To address this issue we have performed additional analysis and experiments:

1. The key extrinsic regulators of the erythroid lineage are Kit ligand/SCF and Epo. We therefore checked if the expression of these cytokines by stromal cells was altered during ageing, and observed no significant differences for any of the cell types analysed. We have added these data as Supplementary Figure 3g,h (The *Kitl* graph will be removed from Supplementary Figure 4 to avoid duplication; *Epo* was not detected in either condition).

2. In addition, we found that short-term exposure of young bone marrow to IL-6 had the opposite effect of anti-IL-6 antibody treatment of aged bone marrow (i.e. increase of GMP and decrease of preMegE populations, with no change to preCFU-Es/CFU-Es) (new Supplementary Figure 3i,j). These results are therefore fully consistent with the proposed contribution of IL-6 to the aged progenitor phenotype.

Both of these observations support a direct effect of IL-6 on erythroid progenitors as the key mechanism of suppression in aged bone marrow, consistent with the observation that culturing

erythroid progenitors in the presence of IL-6 impairs their colony forming capacity *in vitro* (Mei et al., 2018 – cited). However, while these additional data are clearly consistent with a direct effect of IL-6 being a key component of the observed phenotype it is difficult to completely rule out indirect effect of IL-6 on erythropoiesis via stromal cells. We have therefore added this caveat to the discussion, as it is clearly relevant.

B. The authors argue that TGF β acts independently of IL-6, and plays a different role in the ageing of HSCs, LSK and myelo-erythroid progenitor compartments; here, it is made likely that the stroma itself is actively contributing, as the authors found more TGF- β 1 in ageing compared to young stroma. However, it has been well established that TGF β is a major determinant of HSC quiescence, which is neither referred to nor experimentally addressed here. It is conceivable that the effects observed with the TGF β R-inhibitor in Figures 4 and 5 are simply due to a loss of HSC quiescence, which subsequently resulted in enhanced/increased differentiation of the HSCs and downstream progenitors. It is important that the authors adequately address this possibility and, if necessary, adjust their conclusions accordingly.

These are very valid points.

In young mice TGF β produced by megakaryocytes maintains the HSC quiescence Zhao et al., 2014 – cited), and MKs are enriched in the niches of quiescence of Vwf+ platelet/myeloid-biased HSCs (Pinho et al., 2018 - cited). Overall, our results are therefore fully consistent with a model where increased level of TGF β in the bone marrow (as now measured – new Supplementary Figure 4e) contributes to the maintenance of the expanded population of Vwf+ HSCs that we previously observed in aged mice (Grover et al., 2016 – cited).

The observed affect of ALK5 inhibition in aged mice is a decrease in HSC numbers and loss of HSC platelet/myeloid lineage bias. This would be consistent with the expanded pool of platelet-biased HSCs seen in aged mice being depleted due to removal of a key component of their micro-environmental support, leading to loss of quiescence and exhaustion and/or differentiation. We have now measured the effect of ALK5 inhibition on the LT-HSC cell cycle, and we indeed observe a decrease in HSCs in G0, and a corresponding increase in G1 HSCs, consistent with this notion. We have added these new data (new Figure 4d,e). We therefore fully agree that loss of quiescence is a likely mechanism by which the HSC pool is altered through a selective effect of ALK5 inhibition on HSCs with functional platelet/myeloid bias.

We have modified the discussion to include the above considerations, and have also as requested added additional references to the role of TGF β in HSC quiescence (Zhao et al., 2014) and ageing (Sun et al., 2014).

Minor comments:

C. The authors show a lot of GSEA graphs, but it is unclear for the reader which genes are part of the indicated enrichment. For instance, in Figure 2h there is an increase in so-called “preGM-associated genes” in CFU-Es from old mice, but the identity of these genes is not revealed and it is also not clear which genes within these gene-sets are indeed higher or lower. It would be highly valuable if the GSEA analyses that are shown throughout this manuscript would be expanded by including the related genes in a supplemental figure or table.

These are gene sets that have been previously published, and we therefore did not specify the gene lists or their individual regulation. However, we are happy to do so, and have added the relevant gene sets to Supplementary Table 4.

D. In figure 2, the authors describe that purified MPP2, preCFU-Es and CD71+ CFU-Es from aged mice all have decreased erythroid differentiation (Figure 2f). The prediction is that they all show a gain in myeloid differentiation, but this is only shown for the preCFU-E population in Figure 2i. It would be worth to also show the myeloid potential of the MPP2 population, as this would reveal how far up in the hematopoietic hierarchy the switch from erythroid to myeloid occurs.

Only a fairly small number of MPP2 can be retrieved from each mouse, as this is a rare cell type, and we therefore did not analyse the myeloid potential of this population simply because we prioritised the erythroid colony forming assay. We have performed the experiment again, and this time measured MPP2 myeloid potential. We observe a similar increase in CFU-GM activity in aged MPP2 as that seen in the preCFU-E population, consistent with the effect of ageing on erythropoiesis originating in these early progenitors. The data have been added to Figure 2i, and we have modified the text accordingly.

E. In Figure 3g&h, the authors show in a GSEA that the loss of preCFU-E genes upon ageing is prevented by depletion of IL-6. It is important to also show whether the enrichment of preGM genes upon ageing (Fig 2h) is also lost by the IL-6 depletion. This would denote whether IL-6 truly affects a shift in the balance between erythroid and myeloid commitment, or only affects erythroid differentiation.

F. At the bottom of page 5, the authors refer to Suppl Fig 3e, but this does not exist and should be Suppl Fig 4a.

We thank the reviewer for spotting the error in Sup Fig 3e – this has been corrected. We have done the additional GSEA analysis, and it shows a mild decrease in the enrichment of the preGM signature in aged preCFU-Es after anti-IL-6 antibody treatment. However, while the trend is what would be expected the change is not as significant as for the erythroid signature. We have added these data to Supplementary Figure 3, and adjusted the text and discussion accordingly.

Reviewer #2 (Remarks to the Author):

In the manuscript, "Using bone marrow stromal cells as micro-environmental sensors identifies IL-6 and TGF β signaling as regulators of declining erythropoiesis and lymphopoiesis during hematopoietic ageing," Valletta et al. utilized bone marrow (BM) stromal cells to determine consistent differences in cellular signaling across multiple cell types between young and aged BM and find enhanced IL-6 and TGF beta gene programs in aged BM. Next, hematopoietic stem cells (HSC) were assessed functionally by transplantation to determine if age related defects in erythropoiesis and lymphopoiesis are cell-autonomous, and the author observed small non-significant changes in the intrinsic ability for aged HSC to initiate erythropoiesis while declining lymphopoiesis was a cell-autonomous effect. Taken together these data indicated both cell-autonomous and non-cell-autonomous effects on the ability of aged HSC to produce normal lineage output.

Overall, the work shows some important findings relating to cell-extrinsic signaling networks in aging impacting platelet and erythroid output, which are often ignored in favor of myeloid vs. lymphoid lineage alterations only.

Major concerns:

Inflammatory gene signatures in aging HSC are well described in prior work (Chambers, Plos Biology 2007) Here, RNA-seq analysis of stromal cells identified a similarly inflammatory signature that impacts hematopoiesis. Thus, the value of analyzing gene expression in the stroma or calling them 'sensors' is not clear, given gene expression analysis of HSC/HSPC would yield similar results. Value would be added to this analysis if the impact of the inflammatory signature on HSC maintenance function of the stromal cells themselves is addressed. Indeed a larger issue with the work as a whole is that a preliminary level of mechanistic insight with respect to the impact of TGF-beta and/or IL-6 on aged hematopoietic function is provided.

The reviewer raises a valid point. The reasoning behind using the stromal cells as sensors is that we know that the composition of e.g. HSCs (as analysed in the Chambers paper) is altered significantly during ageing (large increase in the proportion of platelet-biased HSCs, corresponding decrease in multi-lineage HSCs), and it therefore becomes difficult to determine if any molecular differences observed are due to changes at the level of the individual cell caused by altered environmental influence, or due to altered composition of the population. Single cell RNAseq could be used; however, due to technical dropout aggregation of cells would be necessary to generate robust pathway analysis.

We hypothesized that stromal cell populations were less prone to this type of skewing, and also that by analysing multiple populations, and focusing on those pathways that were consistently altered, we could avoid issues related to ageing of specific cell types or changes to their composition. We think, by and large, that the results generated justify our approach, which is complementary to analysis of hematopoietic cell types. However, this does not diminish the validity of investigating altered gene expression in e.g. HSCs for this purpose, as both approaches have their caveats, and confirming results by independent approaches therefore useful. We have now added signatures altered in hematopoietic cells during ageing to the Discussion section.

The issue as to whether the inflammatory signatures impact HSC maintenance, this is a possibility. In addition to TGF β 1 (which we functionally validate) other key HSC regulators produced in the bone marrow include Cxcl12, KitL/SCF and Angpt1. Expression of these was shown in Sup Fig 4. We did not observe the expression of any of these to be systematically down-regulated in aged stromal cells – indeed, in addition to *Tgfb1*, also *Cxcl12* expression was increased. Up-regulation of *Cxcl12* could perhaps contribute to the overall increase in HSC number with age; however, we would consider investigating this to be outside the scope of the present manuscript, which deals mainly with HSC lineage bias.

Increased levels of IL-6/TGF (as opposed to increased signaling activity in these pathways due to cell-intrinsic changes) in the BM are not shown. Indeed other previous analysis of aged BM (Ergen et al, Blood 2012) show decreased IL-6 levels in BM fluid by direct interrogation, whereas the study cited by the authors (Mei et al, 2018) only shows protein levels in serum and not BM (indirect measurement of IL-6 was

performed by qRT-PCR in this study). Echoing the point above relating to mechanism, it will be important for the authors to distinguish between the alternative hypotheses of increased levels of these factors in situ in the BM vs. increased signaling activity due to intrinsic deregulations.

It has been shown that IL-6 protein levels are 4-5 fold elevated in bone marrow plasma in aged mice (Henry et al. 2015) – we have now included this citation and made this point more clearly in the Introduction and Discussion.

The paper mentioned (Ergen et al., Blood 2012) does not actually measure IL-6 concentration in BM plasma, but the amount of IL-6 protein normalized to total protein in the sample (Figure 2 of the paper), and it is not clear how this translates to protein concentration. It is also not clear from the legend how many biological or technical replicates were performed. However, the outcome of this normalisation procedure is that the vast majority (10 of 12) of the cytokines measured show a similar 2-3 fold decrease, potentially indicative of a systematic skewing by the sample preparation or assay procedure. We are more confident of the procedure used by Henry et al., as it involves simply measuring the concentration in bone marrow fluid with no normalization involved, and is consistent with the increase in circulating IL-6 in aged mice observed across multiple studies (which the Henry et al. study also confirms).

We have now also measured the TGF β 1 level in bone marrow fluid from young and old mice, observing a significant increase (data added as Sup Fig 4e. The available data are therefore fully consistent with the altered gene expression signatures in aged stromal cells being extrinsically regulated, rather than intrinsically caused by e.g. chronological ageing. We have added this point to the Discussion.

Reviewer #3 (Remarks to the Author):

In this paper from Valletta et al., the authors attempt to find an answer to the molecular features underlying ageing hematopoiesis/aging HSC behavior. The studies try to argue for a role for IL6 and Tgfb signaling in controlling hematopoiesis at two levels (erythroid progenitor cells and HSCs, respectively). The interpretations appear for the most part to be supported by the data provided and statistics (where applied) seem adequate.

The authors first describe the general features of ageing hematopoiesis, focusing on decreased erythropoiesis and enhanced platelet (biogenesis?). They validate previous findings and apply a transgenic mouse model that permit tracing of these two lineages (demonstrating that also this transgenic strain exhibits these well-known age-associated features) (Figure 1). In essence, the authors conclude that while there is a general reduction of the contribution of aged HSCs to multilineage hematopoiesis, this is more pronounced for the lymphoid lineages.

Not sure if there is anything wrong with the resolution in my figures, but I cannot really see any cells in the gates connected to Figure 1b (plots to far right) and Figure 1d (CD11b vs Gr1 plot; no myeloid cells?).

We thank the reviewer for spotting this problem. We now increased the size of the dots representing individual cells in these plots – hopefully this resolves this issue.

Having verified/established these basic features, the authors next investigate myeloerythroid progenitor cells in young and aged mice. They demonstrate an increase in megakaryocyte progenitors and a decrease of early erythroid progenitors (including MPP2s – it could benefit the general reader to explain a bit more why these cells were investigated). Finally, authors conduct some gene expression studies and culture experiments to demonstrate that the myeloid activity of aged preCFU-Es is higher from aged cells. These latter experiments seem to indicate that the myeloid (as opposed to erythroid) activity/cell goes up with age. This raises some concerns as to the way the prospective isolation was made – if f.i. CD55 had been included (positivity) as a marker for candidate preCFU-Es, would the authors obtain the same results?

We have analysed the expression of CD55 on the preCFU-E (and other erythroid progenitor) populations (preMegE, CFU-E). These were all uniformly CD55 positive in both young and old mice. Therefore, the results would not have been altered by the inclusion of CD55 in the phenotypic definition for these populations. As these data are of a mainly technical nature we would propose not to include them in the manuscript, but are happy to do so if requested. We have appended representative plots to this rebuttal (Figure 1) for information. This shows the uniform expression of CD55 on preCFU-Es and CFU-Es in both young and aged mice – preMegEs were ca. 90% CD55+ (panels A and B). Furthermore, we have confirmed the decreased number of preMegEs, preCFU-Es and CFU-Es in aged mice after adding CD55 to their phenotypic definition.

The authors next go on to use the rather clever approach of using stromal/microenvironmental cells as probes for what might underlie the hematopoietic phenotypes of aging. This led to the identification of two candidate signaling pathways; IL6 and Tgfb. Further experimentation is focusing on the neutralisation of the two pathways, followed by re-evaluation of hematopoiesis. From these lines of studies, the conclusions are: 1) that IL6 exerts negative effects of erythropoiesis (rather than having direct effects on HSCs) while 2) Tgfb-inhibition appears to restore the lineage (or rather progenitor) output from aged HSCs (including the overproduction of megakaryocyte progenitors; what about platelets?). They provide some molecular evidence (several types of genome-wide RNA expression experiments) to support their conclusions.

We have now measure the effect of ALK5 inhibition on platelet output, and observe a reduction in peripheral blood platelet count to the level observed in young mice (new Figure 4h).

Finally, the investigators conduct transplantation experiment using purified HSCs from vehicle or Tgfb-inhibited treated mice. They try to demonstrate that this treatment "normalizes" the output from HSCs with enhanced relative output of non-platelet lineages.

It appears that the overall reconstitution of Tgfb-inhibited cells is dramatically lower than that from vehicle-treated. What is going on here? It appears to this reviewer that Tgfb-inhibition is causing an overall decline in HSC function, that perhaps has some selectivity for HSCs with a "myeloid-bias"/"compromised lymphopoiesis". But this is very difficult to conclude if the variation in overall reconstitution is very compromised – f.i. low-level chimeras tend to be skewed towards "lymphoid-biased" cells (which is most likely reflecting more on the longevity of mature lymphoid cells).

We would agree with the interpretation of the reviewer that there is an overall loss of HSC reconstitution after ALK5 inhibitor treatment, and that the data indicate that this preferentially affects platelet-biased HSCs, leading to rebalancing of HSC output. It is important to note, however, that even after ALK5 inhibition the highest output from HSCs is still of platelets, erythrocytes and myeloid cells, which include the two most short-lived lineages (platelets, myeloid cells). This argues against the engrafted HSCs being functionally compromised – actual loss of HSC self-renewal will lead to outright lymphoid bias, as in this case myeloid and platelet reconstitution will rapidly decline (please see Carrelha et al. Nature 2018, (cited) Fig 4d for supporting data).

This would be consistent with the observation that niches for Vwf+ HSCs (which are enriched in platelet-biased HSCs – Sanjuan-Pla et al., 2013 - cited) are found in proximity to megakaryocytes (de Pinho et al. 2019 – cited) and depend on megakaryocyte-derived TGFβ1 for maintenance of quiescence and function: increased levels of TGFβ1 in aged bone marrow, due to increased local production by other stromal cell types, would therefore have the potential to increase the quiescence and functional maintenance of platelet-biased HSCs by providing additional TGFβ1-producing niches.

We have therefore measured the effect of ALK5 inhibition on the cell cycle status of aged HSCs, and found that quiescence of phenotypic LT-HSCs is decreased after inhibitor treatment (new Figure 4d). This would be consistent with a proportion of LT-HSCs being in the process of losing self-renewal capacity due to lack of micro-environmental support. The relative loss of HSC reconstitution may therefore simply reflect that these HSCs still retain their surface phenotype, but have already been functionally compromised. They would therefore be included in the transplanted HSCs, but not produce long-term engraftment.

Overall, these observations are consistent with ALK5 inhibition leading to loss of HSCs quiescence and function, with platelet-biased HSCs being preferentially affected.

We have added these considerations to the discussion section.

Some emphasis is on the "rescue" of B lymphopoiesis (Figure 5e) as a consequence of Tgfb-inhibition, but details on whether there is restoration of B lymphopoiesis in the bone marrow is lacking. It would have been very interesting for the authors to provide some information specifically on IL7-signaling, as compromised IL7 signaling in the past has been proposed to be a major mechanism for the reduced B (and/or T) lymphopoiesis with age.

First of all, in order to strengthen the data regarding the effect of ALK5 inhibition we have now also performed clonal assays of B-cell progenitors. These new data confirm that there is a recovery of B progenitors, here measured as the number of B-clonogenic progenitor cells, to about 75% of the level in young mice after ALK5 inhibitor treatment (as compared to ca. 25% of the young value in untreated mice). These data have been added to Figure 5.

We did, however, not detect any significant change to the IL-7 dose response after ALK5 inhibition – B-cell progenitors from control and ALK5 inhibitor treated mice showed the same (relatively weak) response to IL-7. These data have been added to Figure 5f and Supplementary Figure 4f.

Therefore, the recovery of B-cell progenitor function after ALK5 inhibition is likely to reflect an increased generation of these progenitors (consistent with the HSC/MPP functional and molecular

data), rather than an increased IL-7 response. We have added these considerations to the Discussion section.

A**B****C**
A) Representative flow cytometry plots of CD55+ erythroid progenitor populations (preMEG-E, preCFU-E and CFU-E) from young (top panels) and old mice (bottom panels).

B) Quantification of erythroid progenitor CD55+ fraction (preMegE, preCFU-E and CFU-E) from young and old mice (N=6/condition). Data are from 2 independent experiments.

C) Bar graph showing the number of CD55+ erythroid progenitors (preMegE, preCFU-E and CFU-E) in bone marrow of young and old mice as frequency of LK cells (N=6/condition, 2 independent experiments). Values are mean \pm s.d. * P<0.05; ***P<0.001 (Student's t-test).

REVIEWERS' COMMENTS:

Reviewer #1 (Remarks to the Author):

The authors have adequately addressed the comments of all reviewers, and have considerably improved the manuscript.

Reviewer #2 (Remarks to the Author):

The authors have satisfactorily addressed my concerns.

Reviewer #3 (Remarks to the Author):

In this study, the authors interrogate the basis for the altered hematopoietic stem cell function with age. Using a combinatorial approach of probing stromal cells as molecular sensors for cytokine signalling, they reveal that both Tgfb and IL6 signalling is altered with age. By a series of functional experiments (predominantly by inhibiting the activity of these mitogens) they are able to separate the functional effects of these pathways on overall lineage potential and erythropoiesis, respectively.

The paper has many original lines of experimentation and is likely to be influential in the field. The results are clearly presented, the methods are well described, and they have adequately addressed all the concerns I raised in connection to my review of their original submitted manuscript.

Response to reviewers' comments:

No additional comments were received.